# Entropy Never Lies: Signed Entropy Integral Unmasks Mislabeled Data

## Abstract

Mislabeled samples in training datasets severely degrade the performance of deep networks, as overparameterized models tend to memorize erroneous labels. We address this challenge by proposing a novel approach for mislabeled data detection that leverages training dynamics. Our method is grounded in the key observation that correctly labeled samples exhibit consistent entropy decrease during training, while mislabeled samples maintain relatively high entropy throughout the training process. Building on this insight, we introduce a signed entropy integral (SEI) statistic that captures both the magnitude and temporal trend of prediction entropy across training epochs. SEI is broadly applicable to classification networks and demonstrates particular effectiveness when integrated with contrastive language-image pretraining (CLIP) architectures. Through extensive experiments on three medical imaging datasets—a domain particularly susceptible to labeling errors due to diagnostic complexity—spanning diverse modalities and pathologies, we demonstrate that SEI achieves state-of-the-art performance in mislabeled data identification, outperforming existing methods while maintaining computational efficiency and implementation simplicity.

## 1 Introduction

Deep networks have achieved remarkable results in medical imaging, enabling applications from tumor segmentation to disease classification (Litjens et al., 2017). Yet, their performance hinges on the quality of training data (Shi et al., 2024). In practice, medical datasets often contain mislabeled samples due to the complexity of diagnosis, inter-observer variability, and limited annotation resources (Shi et al., 2024; Alderman et al., 2025). For instance, in dermatology, the visual appearance of skin lesions can overlap heavily between malignant and benign conditions; melanomas may resemble benign nevi in early stages, and fungal infections can mimic inflammatory skin disorders. Even experienced dermatologists may disagree without histopathological confirmation. Similarly, in ophthalmology, subtle retinal changes in early diabetic retinopathy can be challenging to grade consistently, especially when annotation guidelines differ across graders. Such challenges mean that noisy labels are a realistic concern in many medical imaging datasets.

Noisy labels pose a particular risk for overparameterized deep networks, which can fit even randomly assigned labels given sufficient capacity (Zhang et al., 2017). When this happens, a model effectively memorizes mislabeled samples by learning overly specific, non-generalizable features. For example, if an image of a benign skin nevus is mistakenly labeled as melanoma, the model may latch onto spurious visual patterns unique to that single image—such as lighting artifacts or sensor noise—rather than features indicative of melanoma. These spurious correlations will lead to overfitting and degraded performance.

Our aim is to automatically identify and remove mislabeled samples from training data. This not only uncovers systematic annotation errors but also improves overall label quality. Moreover, because large medical datasets are often too extensive for exhaustive manual inspection, automated methods are essential for isolating mislabeled samples without overburdening domain experts.

Prior works on mislabeled data detection typically involve multi-stage pipelines and tailored loss functions or modules (Chen et al., 2019; 2024; Huang et al., 2019; Li et al., 2020; Cheng et al., 2021; Wei et al., 2024), which can be tightly coupled to specific architectures or disrupt standard training workflows. We take a different approach: a simple yet effective plug-and-play method that works

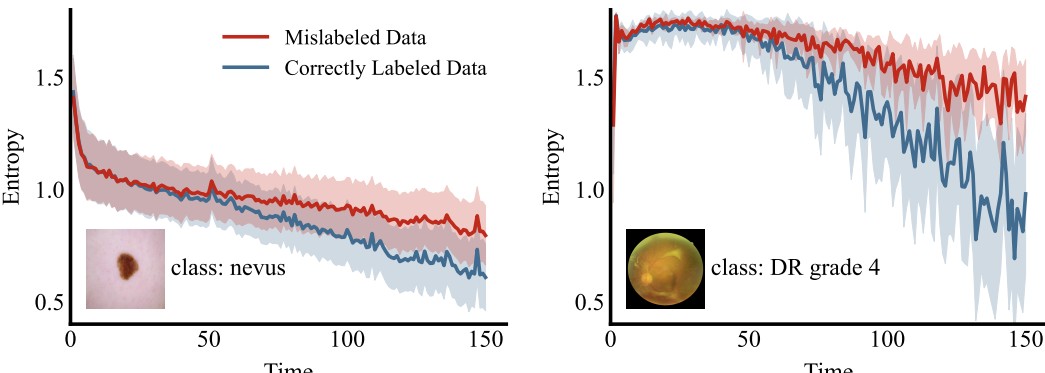

Figure 1: Training dynamics of prediction entropy for correctly labeled versus mislabeled samples. Left: Nevus images from the ISIC dataset, comparing correctly labeled samples (ground truth: nevus; given label: nevus) with mislabeled samples (ground truth: nevus; given label: other skin lesion categories). Right: Grade 4 diabetic retinopathy (DR) images from the DeepDRiD dataset. In both cases, correctly labeled samples exhibit steadily decreasing entropy throughout training, while mislabeled samples maintain persistently high entropy, demonstrating the potential of entropy as an indicator for noisy label detection.

with existing training setups and requires minimal changes. Our method tracks training dynamics of a classification model with a contrastive loss (Radford et al., 2021) to align medical images and labels. We observe that the evolution of prediction entropy encodes subtle cues for distinguishing noisy from clean samples. Correctly labeled samples tend to exhibit a steady entropy decrease as learning progresses, whereas mislabeled samples often maintain relatively high entropy.

However, entropy alone is insufficient to separate mislabeled data from hard but valuable clean samples, which may also retain high entropy despite having correct labels. To address this, we propose a signed entropy integral (SEI) statistic that captures both the magnitude and trend of entropy evolution. This signed formulation provides a richer characterization of training dynamics: hard clean samples and mislabeled samples exhibit distinct patterns in how their entropy changes over time, allowing SEI to differentiate the two.

Our work makes three key contributions:

- Through large-scale analysis of training dynamics, we identify two informative signals for distinguishing mislabeled from correctly labeled samples: entropy evolution and label-prediction consistency over time.

- Building on these insights, we introduce *signed entropy*, a novel extension of Shannon entropy that incorporates label consistency, and propose the SEI statistic that captures the cumulative training behavior of different sample types.

- we demonstrate that our simple yet effective method achieves state-of-the-art performance on three medical imaging datasets spanning different modalities and pathologies, without requiring architectural modifications or complex training procedures.

## 2 METHODOLOGY

### 2.1 PROBLEM FORMULATION

We consider the task of $K$-class image classification, where the objective is to train a model that predicts a label $y$ for an input image $x$. The training dataset $\mathcal{D}_{\texttt{train}} = \{(x_i, y_i)\}$ contains two types of samples. A *mislabeled* sample is one whose assigned label does not match its underlying semantic content (e.g., $x$ is an image of a melanoma but the label $y$ is "nevus"). A *correctly labeled* sample is one where the assigned label aligns with the true category. Our goal is to identify mislabeled data in $\mathcal{D}_{\texttt{train}}$ by exploiting differences in their training dynamics compared to correctly labeled data.

## 2.2 Preliminary: CLIP for Image Classification

CLIP (Radford et al., 2021) demonstrates strong zero-shot performance by learning joint visual-textual representations from large-scale image-text pairs using a contrastive objective. Classification is performed by measuring the similarity between image features and text embeddings derived from prompts such as "a photo of [CLS]", where [CLS] denotes a class name. The prediction probability is computed as

$$p(y = k|\boldsymbol{x}) = \frac{\exp(\text{sim}(\boldsymbol{v}, \boldsymbol{t}_k)/\tau)}{\sum_{j=1}^{K} \exp(\text{sim}(\boldsymbol{v}, \boldsymbol{t}_j)/\tau)} \,, \tag{1}$$

where $\text{sim}(\boldsymbol{v}, \boldsymbol{t}_k)$ denotes the cosine similarity between image feature $\boldsymbol{v}$ and text embedding $\boldsymbol{t}_k$, and $\tau$ is a temperature parameter. In this framework, each image is compared against all $K$ class-specific text embeddings, allowing us to incorporate dataset labels directly while preserving the representational benefits of visual-textual alignment.

## 2.3 Findings

To understand training dynamics of mislabeled samples, we first examine their prediction *entropy*. Figure 1 shows entropy trajectories for nevus images in the ISIC dataset and grade 4 diabetic retinopathy images in the DeepDRiD dataset (Liu et al., 2022), comparing correctly labeled samples with mislabeled ones. Early in training, the two groups are entangled, exhibiting similar levels of uncertainty. As training progresses, however, a clear separation emerges: entropy for correctly labeled data decreases steadily, while mislabeled samples maintain elevated entropy. This suggests that entropy provides a promising signal for distinguishing mislabeled samples from clean ones. We observe similar trends across other classes and datasets (see Appendix C).

Nevertheless, entropy alone proves insufficient for reliable detection. Figure 2 illustrates this limitation by comparing a mislabeled sample with a hard-to-learn but correctly labeled sample. Despite their fundamentally different ground-truth status, their entropy curves are nearly indistinguishable, making it difficult to separate the two based solely on this statistic.

To address this limitation, we investigate an additional aspect of training dynamics: the *alignment between given labels and model predictions over time*. Figure 3 presents alignment statistics for three distinct sample categories: easy clean, hard clean, and mislabeled data, where circle size and opacity reflect frequency. We observe that: (1) most easy clean samples exhibit consistent alignment, with predictions typically matching their labels throughout training; (2) hard clean samples generally demonstrate mixed dynamics; (3) mislabeled samples are predominantly characterized by persistent misalignment with their given labels. These distinctive alignment patterns provide complementary information beyond entropy, enabling better discrimination between mislabeled samples and challenging yet correctly labeled ones.

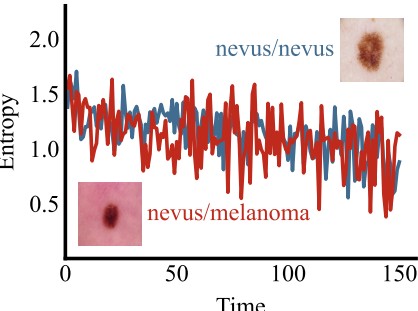

Figure 2: Entropy trajectories for a mislabeled sample (ground truth: nevus; given label: melanoma) and a hard clean sample (ground truth: nevus; given label: nevus). Despite differing label correctness, their entropy curves are nearly indistinguishable. This illustrates that entropy alone cannot reliably distinguish mislabeled data from challenging but clean examples.

## 2.4 Identifying Mislabeled Data

### 2.4.1 Signed Entropy

The observations in Section 2.3 suggest that two complementary cues—entropy dynamics and label-prediction consistency—can be jointly exploited for identifying mislabeled data. We therefore introduce *signed entropy*, a quantity that extends Shannon entropy with a label consistency term.

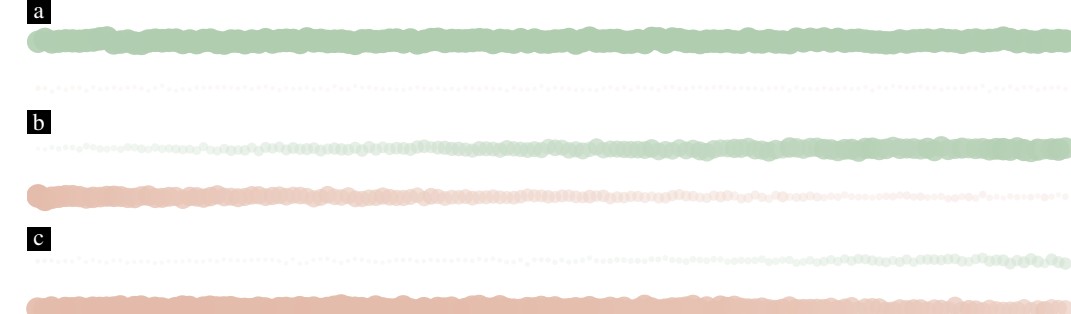

Figure 3: Training dynamics analysis through label-prediction alignment patterns over time. Bubble timeline charts illustrate the evolution of three sample categories during training: (a) easy clean samples, (b) hard clean samples, and (c) mislabeled samples. At each time step, ● green circles indicate alignment between predicted and given labels, while ● red circles denote misalignment. Circle size and opacity encode frequency.

Formally, for any $(\boldsymbol{x}, y) \in \mathcal{D}_{\texttt{train}}$, let $\boldsymbol{p}(\boldsymbol{x}) = (p_1(\boldsymbol{x}), \ldots, p_K(\boldsymbol{x}))$ denote the model's posterior distribution over $K$ classes (cf. Eq. 1). We define signed entropy as

$$\mathcal{H}(\boldsymbol{p}(\boldsymbol{x}), y) = (-1)^{\mathbb{1}\left[y = \arg\max_k p_k(\boldsymbol{x})\right]} \sum_{k=1}^{K} p_k(\boldsymbol{x}) \log p_k(\boldsymbol{x}), \tag{2}$$

where the exponent introduces a sign depending on whether the assigned label $y$ matches the model's prediction. In other words, $\mathcal{H}$ reduces to Shannon entropy when $y$ agrees with the prediction, but flips its sign under misalignment.

**Discussion**    Shannon's entropy is always nonnegative and only reflects distributional uncertainty, making it blind to label correctness: both hard clean samples and mislabeled ones may exhibit high entropy. In contrast, the signed entropy in Eq. 2 attaches a directionality:

- **Positive signed entropy** indicates both uncertainty and label consistency, as typically seen in clean samples (easy or hard).
- **Negative signed entropy** emerges when the model contradicts the given label, flagging potential annotation errors.

### 2.4.2    Signed Entropy Integral

While signed entropy at a single epoch can provide useful information, training dynamics often fluctuate, making individual snapshots unreliable for detecting mislabeled samples. Moreover, it is unclear a priori which epochs contain the most discriminative signals. Aggregating *cumulative behavior* across training offers a more stable criterion, as it naturally smooths out such fluctuations. To this end, we introduce SEI, which accumulates signed entropy values over the entire training trajectory.

Let $\boldsymbol{p}^{(t)}(\boldsymbol{x})$ denote the posterior distribution at epoch $t$. SEI is defined as

$$\text{SEI}(\boldsymbol{x}, y) = \sum_{t=1}^{T} \mathcal{H}(\boldsymbol{p}^{(t)}(\boldsymbol{x}), y), \tag{3}$$

where $T$ is the total number of training epochs.

**Analysis**    Figure 4 illustrates how SEI separates different sample types. For an easy clean sample, the signed entropy remains mostly positive, as predictions stay consistent with its assigned label, resulting in a large positive integral. A hard clean sample may initially exhibit misalignment and accumulate negative contributions, but as training progresses, the signed entropy turns positive; the resulting positive and negative areas partly cancel, leading to a smaller integral. By contrast, a

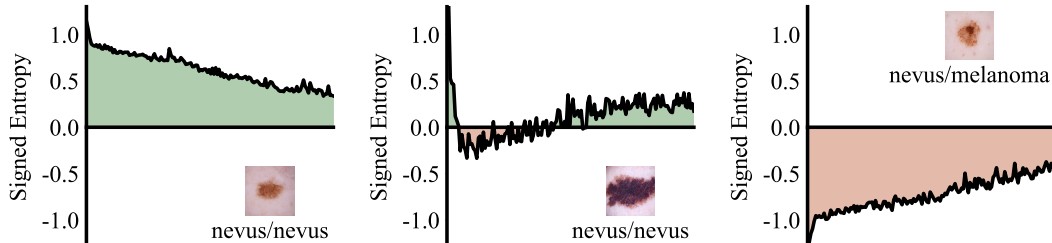

Figure 4: Illustration of SEI. The plots depict signed entropy curves over training epochs for easy clean (left), hard clean (middle), and mislabeled (right) samples. Each curve is averaged over 200 samples, and the signed area under each curve corresponds to the SEI. Correctly labeled samples yield larger SEIs than mislabeled ones.

mislabeled sample typically shows persistent disagreement, so its signed entropy curve stays negative throughout training, yielding a strongly negative integral.

In this way, SEI produces a single scalar that naturally ranks samples: mislabeled ones cluster at the negative end, while correctly labeled ones occupy positive values. This simple measure proves effective across datasets and modalities for isolating annotation errors from both easy and hard clean samples (see Appendix D).

## 2.5 Thresholding

We require a threshold to distinguish between clean and mislabeled data. A fixed threshold is impractical since the distribution of SEI values varies across datasets and training configurations. Instead, we use an adaptive thresholding strategy that leverages artificially mislabeled samples as reference points.

Concretely, given $N$ training samples across $K$ classes, we randomly select $N/(K+1)$ instances and reassign their labels to an auxiliary class $K+1$ that does not exist in the dataset. In the CLIP setting, this corresponds to using a synthetic prompt such as "a dermoscopic image showing other lesions". Since the auxiliary class is designed to be semantically meaningless, these relabeled samples serve as natural surrogates for mislabeled data. Moreover, choosing $N/(K+1)$ instances ensures that the auxiliary class appears with a frequency comparable to the original classes, preventing imbalance in calibration.

We then compute SEIs for all auxiliary-class samples and use their average value as the decision threshold. Any sample whose SEI falls below this threshold is flagged as mislabeled. This simple scheme yields an adaptive, data-driven cutoff without requiring manual tuning.

## 3 Related Work

Addressing noisy labels in datasets has spawned two primary research directions: (1) Explicit mislabeled data detection aims to identify and remove incorrectly labeled instances to improve data quality, while (2) Noise robust learning develops algorithms that maintain performance despite label noise.

**Mislabeled Data Detection**  Most methods exploit training signals as proxies for label correctness. Loss-based approaches leverage the intuition that higher losses often indicate incorrect labels. O2U-Net (Huang et al., 2019) alternates between overfitting and underfitting phases, identifying mislabeled samples through consistently higher normalized losses. CORES (Cheng et al., 2021) progressively filters incorrectly labeled instances using training loss in a proposed sample sieve framework. Beyond loss, various proxy measures have been developed, including gradient-based metrics (Zhang & Sabuncu, 2018) and prediction-based statistics (Northcutt et al., 2021; Pleiss et al., 2020; Song et al., 2019). For instance, Northcutt et al. (2021) filter low-confidence samples using class-conditional thresholds on predicted probabilities.

Recent work has explored training-free approaches. SIMIFEAT (Zhu et al., 2022) detects noisy labels through k-nearest neighbors in the feature space of a pre-trained model, while DEFT (Wei et al., 2024) leverages CLIP's image-text alignment to learn class-specific prompts for mislabel detection. LEMoN (Zhang et al., 2025) exploits multimodal neighborhoods of image-caption pairs in the latent space of CLIP to automatically identify label errors. However, the effectiveness of these methods depends heavily on the generalization capacity of pre-trained models, which may be limited in specialized domains like medical imaging. ReCoV (Chen et al., 2024) identifies mislabeled medical data through cross-validation, though this increases computational overhead for large datasets. Some approaches (Wang et al., 2024) assume access to clean subsets, while we consider the more restrictive but realistic setting where no training data can be trusted.

**Noise Robust Learning** Rather than targeting specific noisy instances, robust learning methods design modules enabling effective training on noisy datasets. This includes novel architectures (Cheng et al., 2020; Xiao et al., 2015), loss functions (Wang et al., 2019; Ye et al., 2023), regularization techniques (Cheng et al., 2022; Liu et al., 2020), and training strategies (Lukasik et al., 2020; Xia et al., 2021; Yuan et al., 2024). Some works integrate noisy label detection into training pipelines through loss re-weighting (Ren et al., 2018; Bae et al., 2024) or re-annotation (Han et al., 2019; Arazo et al., 2019; Englesson & Azizpour, 2024). Our work focuses on identifying reliably labeled data rather than recycling mislabeled samples. For simplicity, we discard instances flagged as mislabeled; nevertheless, our method is compatible with approaches that attempt to reuse them (cf. Section 4.3.4).

## 4 EXPERIMENTS

We evaluate our approach through two complementary assessments: first, we measure mislabeled data detection performance on synthetically corrupted datasets to directly assess identification capability; second, we evaluate downstream classification performance.

### 4.1 EXPERIMENTAL SETUP

#### 4.1.1 DATASETS

We conduct experiments on three medical imaging datasets spanning different modalities and diagnostic tasks.

**ISIC 2018.** We use the ISIC 2018 Challenge[1] Task 3 dataset for skin lesion diagnosis, containing dermoscopic images across seven categories: melanoma, nevus, basal cell carcinoma, actinic keratosis/intraepithelial carcinoma, benign keratosis, dermatofibroma, and vascular lesion. The dataset comprises 10,015 training images and 1,512 test images.

**DeepDRiD.** This fundus photography dataset (Liu et al., 2022) targets diabetic retinopathy severity grading using a 5-point scale (0-4) following the International Clinical Diabetic Retinopathy standard. Each patient contributes dual-view images (macula-centered and optic disc-centered) for both eyes. We use 1,200 training images and 400 test images from the official split.

**PANDA.** This dataset[2] contains 10,616 whole slide images (WSIs) of digitized H&E-stained prostate biopsies from Radboud University Medical Center and Karolinska Institute. We focus on the Radboud subset, which provides pixel-level annotations distinguishing background, stroma, benign epithelium, and cancerous epithelium (subdivided into Gleason patterns 3, 4, and 5). Since Gleason grading depends solely on epithelial architecture, we consider four classes: benign epithelium, Gleason 3, Gleason 4, and Gleason 5. Each WSI is tiled into $224 \times 224$ patches, with background-dominated patches discarded. For remaining patches, we compute area proportions of epithelial categories from pixel-level masks and apply a dominant-label rule: patches are assigned the highest-grade category present (priority: Gleason 5 > Gleason 4 > Gleason 3 > benign epithelium) if it covers more than 80% of non-background pixels; otherwise, patches are discarded. This yields 4,102 benign epithelium patches and 6,914, 6,413, and 6,956 patches for Gleason grades 3, 4, and 5, respectively. We use a 4:1 train/test split.

---

[1] https://challenge.isic-archive.com/data/#2018
[2] https://kaggle.com/competitions/prostate-cancer-grade-assessment

Table 1: Comparison with state-of-the-art mislabeled data detection methods under symmetric noise. Results are reported in F1 score (%) across five noise rates ($\eta \in \{0.1, 0.2, 0.3, 0.4, 0.5\}$) on three medical datasets (ISIC, DeepDRiD, and PANDA). The best results for each dataset and noise rate are highlighted in **bold**.

| | ISIC | | | | | DeepDRiD | | | | | PANDA | | | | |
| | 0.1 | 0.2 | 0.3 | 0.4 | 0.5 | 0.1 | 0.2 | 0.3 | 0.4 | 0.5 | 0.1 | 0.2 | 0.3 | 0.4 | 0.5 |
|---|---|---|---|---|---|---|---|---|---|---|---|---|---|---|---|
| INCV | 38.60 | 42.72 | 47.21 | 55.93 | 61.67 | 33.05 | 44.56 | 49.29 | 57.87 | 63.32 | 53.30 | 57.69 | 61.23 | 68.45 | 67.11 |
| BMM | 30.49 | 39.59 | 40.65 | 55.89 | 55.73 | 30.23 | 39.44 | 46.11 | 50.66 | 56.21 | 54.37 | 62.88 | 66.19 | 71.09 | 72.43 |
| GMM | 36.31 | 47.49 | 49.67 | 64.79 | 67.10 | 35.89 | 45.39 | 52.28 | 58.87 | 66.27 | 59.60 | 62.79 | 68.24 | 71.11 | 70.76 |
| AUM | 48.65 | 62.97 | 72.60 | 77.70 | 81.67 | 39.66 | 54.96 | 60.92 | 69.20 | 75.75 | 65.23 | 72.39 | 75.95 | 75.30 | 77.18 |
| CORES | 36.20 | 56.90 | 67.84 | 76.72 | 82.67 | 26.21 | 42.68 | 56.38 | 69.22 | 68.26 | 60.73 | 63.61 | 66.28 | 73.43 | 70.51 |
| CL | 34.15 | 39.98 | 43.53 | 44.55 | 43.56 | 39.59 | 47.12 | 53.84 | 51.28 | 62.17 | 54.81 | 59.87 | 62.40 | 69.09 | 72.30 |
| SIMIFEAT | 32.01 | 39.33 | 43.95 | 44.82 | 41.62 | 36.04 | 45.72 | 53.67 | 55.06 | 61.53 | 55.82 | 61.15 | 68.12 | 70.56 | 69.76 |
| DEFT | 25.67 | 38.06 | 44.01 | 52.78 | 55.40 | 29.84 | 42.36 | 45.14 | 52.08 | 62.95 | 53.69 | 59.87 | 62.25 | 67.99 | 69.59 |
| ReCoV | 42.15 | 46.78 | 52.90 | 61.12 | 64.31 | 37.09 | 50.94 | 54.75 | 57.69 | 63.30 | 69.24 | 68.84 | 73.06 | 72.09 | 72.49 |
| LEMoN | 38.41 | 55.86 | 61.97 | 73.70 | 75.35 | 31.82 | 41.64 | 48.20 | 64.35 | 69.55 | 59.60 | 62.54 | 73.95 | 74.30 | 75.58 |
| Ours | **50.44** | **65.64** | **74.80** | **80.07** | **83.93** | **45.20** | **56.53** | **63.43** | **71.26** | **78.19** | **72.57** | **77.22** | **81.46** | **83.11** | **81.18** |

### 4.1.2 SYNTHETIC NOISE GENERATION

To simulate real-world annotation errors, we synthesize mislabeled samples at five noise rates $\eta \in \{0.1, 0.2, 0.3, 0.4, 0.5\}$ using two corruption strategies:

**Symmetric Noise.** In a $K$-class setting, each sample with ground-truth label $a \in \{1, \ldots, K\}$ retains its correct label with probability $1 - \eta$. With probability $\eta$, it is corrupted to a different class $b \neq a$, chosen uniformly from the $K - 1$ alternatives:

$$p_{a \to b} = \frac{\eta}{K - 1}, \quad b \neq a.$$ (4)

**Confusion-Calibrated Noise.** To simulate realistic errors, we first train a reference ResNet-50 classifier and compute an empirical confusion matrix $T \in \mathbb{R}^{K \times K}$ from its predictions. For a sample with ground-truth label $a$, the label remains unchanged with probability $1 - \eta$. With probability $\eta$, it is corrupted to class $b$ ($b \neq a$) according to:

$$p_{a \to b} = \eta \frac{\exp(T_{ab})}{\sum_{k \neq a} \exp(T_{ak})}, \quad b \neq a.$$ (5)

This preserves the target corruption rate $\eta$ while aligning errors with observed class confusions.

### 4.1.3 EVALUATION METRICS

For mislabeled data detection, we use F1 score as our primary metric, following prior works (Zhu et al., 2022; Kim et al., 2024). F1 score balances precision and recall, providing a more reliable assessment. For downstream image classification, we report accuracy, F1 score, and AUC for comprehensive evaluation.

### 4.1.4 IMPLEMENTATION DETAILS

All experiments use PyTorch and run on a single NVIDIA RTX 4090 GPU. For mislabeled data identification, we employ CLIP with a Transformer text encoder and ResNet-50 vision encoder. Training uses SGD with momentum 0.9, weight decay $1 \times 10^{-4}$, batch size 128, and initial learning rate $1 \times 10^{-3}$ for 150 epochs. The learning rate decays by 0.1 at epochs 75 and 115. Images are resized to $224 \times 224$ and normalized using ImageNet statistics (Russakovsky et al., 2015). For downstream classification tasks, we use the official dataset splits.

### 4.2 COMPARISON WITH STATE-OF-THE-ART METHODS

We evaluate our approach against existing mislabeled data indentification methods, including INCV (Chen et al., 2019), BMM (Arazo et al., 2019), GMM (Li et al., 2020), AUM (Pleiss et al., 2020), CORES (Cheng et al., 2021), CL (Northcutt et al., 2021), SIMIFEAT (Zhu et al., 2022), DeFT

Table 2: Comparison with state-of-the-art mislabeled data detection methods under confusion-calibrated noise.

| | ISIC | | | | | DeepDRiD | | | | | PANDA | | | | |
|---|---|---|---|---|---|---|---|---|---|---|---|---|---|---|---|
| | 0.1 | 0.2 | 0.3 | 0.4 | 0.5 | 0.1 | 0.2 | 0.3 | 0.4 | 0.5 | 0.1 | 0.2 | 0.3 | 0.4 | 0.5 |
| INCV | 32.99 | 40.44 | 44.03 | 51.62 | 56.64 | 33.15 | 43.50 | 46.54 | 48.60 | 50.68 | 41.19 | 50.44 | 59.96 | 63.54 | 66.90 |
| BMM | 37.01 | 26.92 | 36.10 | 57.36 | 52.22 | 31.50 | 36.63 | 45.24 | 50.18 | 52.09 | 46.59 | 63.23 | 55.36 | 60.51 | 65.10 |
| GMM | 31.17 | 36.38 | 42.14 | 48.94 | 59.09 | 37.82 | 47.33 | 54.34 | 56.46 | 57.72 | 54.15 | 58.11 | 64.58 | 67.93 | 69.24 |
| AUM | 42.96 | 57.24 | 65.96 | 64.54 | 71.42 | 41.35 | 48.21 | 56.47 | 60.55 | 61.84 | 61.30 | 68.29 | 72.31 | 76.08 | 76.52 |
| CORES | 35.75 | 55.64 | 65.56 | 62.49 | 42.87 | 26.59 | 39.93 | 44.01 | 36.17 | 47.67 | 45.66 | 54.72 | 60.25 | 63.09 | 65.64 |
| CL | 38.14 | 43.44 | 48.66 | 45.46 | 43.28 | 34.06 | 37.60 | 43.06 | 42.78 | 51.58 | 51.87 | 60.77 | 69.19 | 71.53 | 72.35 |
| SIMIFEAT | 38.54 | 48.21 | 49.66 | 50.65 | 46.17 | 32.13 | 40.98 | 41.57 | 42.38 | 48.57 | 59.97 | 61.08 | 68.25 | 69.10 | 71.94 |
| DEFT | 15.33 | 24.67 | 32.72 | 37.03 | 34.41 | 18.46 | 28.74 | 39.68 | 46.53 | 50.34 | 40.66 | 54.72 | 65.48 | 67.83 | 69.93 |
| ReCoV | 40.90 | 44.21 | 61.16 | 64.66 | 66.61 | 42.63 | 48.49 | 50.93 | 52.67 | 61.79 | 59.50 | 63.15 | 64.58 | 65.91 | 69.87 |
| LEMoN | 30.59 | 42.50 | 56.70 | 61.91 | 67.07 | 34.06 | 42.62 | 55.80 | 63.86 | 66.82 | 56.40 | 66.66 | 70.22 | 74.80 | 75.13 |
| Ours | **47.75** | **59.35** | **67.25** | **74.98** | **79.91** | **46.59** | **52.65** | **62.84** | **68.35** | **73.04** | **73.17** | **78.21** | **81.86** | **81.96** | **81.85** |

Table 3: Ablation study on the effectiveness of the signed term and temporal integration. EI denotes the unsigned entropy integral (standard Shannon entropy), SE@T represents signed entropy at the final epoch, SE@T/2 denotes signed entropy at mid-training, and SEI is our full signed entropy integral method.

| | ISIC | | | | | DeepDRiD | | | | | PANDA | | | | |
|---|---|---|---|---|---|---|---|---|---|---|---|---|---|---|---|
| | 0.1 | 0.2 | 0.3 | 0.4 | 0.5 | 0.1 | 0.2 | 0.3 | 0.4 | 0.5 | 0.1 | 0.2 | 0.3 | 0.4 | 0.5 |
| EI | 35.31 | 45.05 | 50.46 | 60.77 | 63.65 | 42.91 | 47.86 | 53.85 | 59.28 | 61.14 | 51.76 | 61.50 | 64.26 | 67.26 | 65.86 |
| SE@T | 35.73 | 44.64 | 52.01 | 57.89 | 63.20 | 39.16 | 44.33 | 51.96 | 57.93 | 61.29 | 41.69 | 56.26 | 62.27 | 62.91 | 56.85 |
| SE@T/2 | 36.97 | 48.19 | 56.30 | 63.72 | 68.73 | 40.32 | 48.87 | 54.89 | 62.84 | 66.58 | 57.35 | 60.06 | 64.80 | 66.26 | 66.88 |
| SEI | **47.75** | **59.35** | **67.25** | **74.98** | **79.91** | **46.59** | **52.65** | **62.84** | **68.35** | **73.04** | **73.17** | **78.21** | **81.86** | **81.96** | **81.85** |

(Wei et al., 2024), ReCoV (Chen et al., 2024), and LEMoN (Zhang et al., 2025). As summarized in Tabels 1 and 2, our approach consistently outperforms all competing methods across all datasets and noise levels.

Under the more challenging confusion-calibrated noise setting, our method delivers significant improvements in mislabeled data detection. On the ISIC dataset, it outperforms the second-best approach by 4.79%, 2.11%, 1.29%, 10.32%, and 8.49% across different noise rates. On DeepDRiD, the gains are 3.96%, 4.16%, 6.37%, 4.49%, and 6.22%, while on PANDA, our method achieves improvements of 11.87%, 9.92%, 9.55%, 5.88%, and 5.33%. Under the symmetric noise setting, we again observe consistent and notable performance boosts, further confirming the effectiveness of our approach.

### 4.3 ANALYSIS AND DISCUSSION

#### 4.3.1 EFFECTIVENESS OF THE SIGNED TERM

To assess the contribution of the signed component in Eq. 2, we compare SEI against an unsigned counterpart using standard Shannon entropy. As reported in Table 3, SEI consistently outperforms the unsigned variant, validating the effectiveness of signed entropy.

We further visualize score distributions for clean and mislabeled samples under both formulations in Appendix F.1.

#### 4.3.2 EFFECTIVENESS OF TEMPORAL INTEGRATION

To evaluate the integral component, we compare SEI with two single-epoch baselines: SE@T (signed entropy at the final epoch) and SE@T/2 (signed entropy at mid-training). Table 3 demonstrates that SEI achieves superior F1 scores, indicating that single snapshots provide unreliable signals while temporal integration yields robust detection.

The integral accumulates directional evidence over time, with each epoch contributing a signed cue: positive when predictions align with assigned labels, negative otherwise. Mislabeled samples accumulate predominantly negative values, while hard clean samples eventually offset early negative contributions through later positive ones. Single-epoch measurements suffer from training fluctu-

Table 4: Architecture generalizability of SEI. We compare performance using standard classification networks (ResNet-50, ViT-B/16) and CLIP with different visual backbones (ResNet-50, ViT-B/16). Results demonstrate that SEI remains effective across diverse architectures.

| | ISIC | | | | | DeepDRiD | | | | | PANDA | | | | |
|---|---|---|---|---|---|---|---|---|---|---|---|---|---|---|---|
| | 0.1 | 0.2 | 0.3 | 0.4 | 0.5 | 0.1 | 0.2 | 0.3 | 0.4 | 0.5 | 0.1 | 0.2 | 0.3 | 0.4 | 0.5 |
| RNet-50 | 45.68 | 57.58 | 65.07 | 73.04 | 76.33 | 43.99 | 51.34 | 61.17 | 66.89 | 71.75 | 71.21 | 75.64 | 79.21 | 79.28 | 78.25 |
| ViT-B/16 | 44.25 | 56.71 | 64.57 | 72.12 | 75.91 | 42.88 | 50.47 | 60.71 | 65.68 | 71.31 | 69.44 | 74.17 | 78.89 | 78.86 | 76.03 |
| CLIP$_{RNet-50}$ | **47.75** | **59.35** | **67.25** | **74.98** | **79.91** | **46.59** | **52.65** | **62.84** | **68.35** | **73.04** | **73.17** | **78.21** | **81.86** | **81.96** | **81.85** |
| CLIP$_{ViT-B/16}$ | 46.66 | 58.36 | 66.89 | 73.59 | 77.25 | 44.58 | 51.78 | 61.27 | 67.74 | 72.41 | 71.92 | 76.52 | 80.63 | 81.28 | 79.45 |

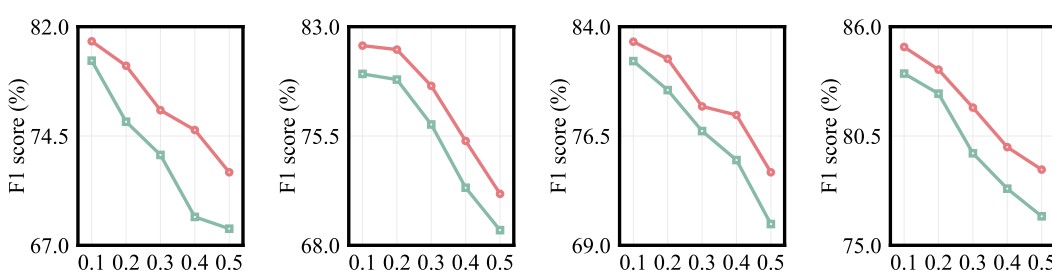

Figure 5: F1 score comparison between baseline noisy label learning methods (green) and their SEI-enhanced variants (red). From left to right: SCE, M-correction, DivideMix, and ProMix.

ations, which temporal integration effectively smooths. This enables the integral to better capture long-term consistency patterns and reduces false detection of hard clean samples.

Besides, we analyze the timing of temporal evidence in Appendix F.2, comparing windowed integrals: SEI@Early (epochs 1–75) and SEI@Late (epochs 76–150).

### 4.3.3 ARCHITECTURE GENERALIZABILITY

To assess the broader applicability of our approach beyond CLIP, we evaluate SEI with standard classification networks including ResNet-50 (He et al., 2016) and ViT (Dosovitskiy et al., 2021). Table 4 shows that while performance decreases, results remain competitive. This demonstrates that: (1) strong mislabeled data detection performance stems primarily from our proposed SEI rather than the CLIP architecture itself; (2) CLIP nevertheless provides advantages, likely due to its contrastive learning objective.

We also evaluate CLIP with various vision encoder backbone to assess generalization across both CNN and Transformer architectures, confirming consistent performance.

### 4.3.4 SYNERGY BETWEEN SEI AND LEARNING WITH NOISY LABELS

To further demonstrate the utility of our approach, we integrate SEI as a data cleaning module with four representative noisy label learning methods: SCE (Wang et al., 2019), M-correction (Arazo et al., 2019), DivideMix (Li et al., 2020), and ProMix (Xiao et al., 2023). The resulting variants, SCE+SEI, M-correction+SEI, DivideMix+SEI, and ProMix+SEI are evaluated against their respective baselines on the ISIC dataset under confusion-calibrated noise. As shown in Figure 5, incorporating SEI consistently improves performance in F1 score (results on accuracy and AUC are provided in Appendix G). These gains highlight the plug-and-play nature of SEI: it is architecture-agnostic and integrates seamlessly into diverse noisy label learning frameworks.

## 5 CONCLUSION

We present SEI, a simple yet effective metric for detecting mislabeled data by leveraging signed entropy dynamics during training, which integrates seamlessly into standard training workflows. Extensive experiments on diverse medical imaging datasets demonstrate that SEI achieves state-of-the-art performance while remaining efficient and easy to apply.

ETHICS STATEMENT

The authors acknowledge that this work adheres to the ICLR Code of Ethics.

REPRODUCIBILITY STATEMENT

Code to reproduce all experiments is available at `https://anonymous.4open.science/r/SEI-03E8/`.

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

Appendix

## A    Use of Large Language Models

Large language models were used solely for light editing tasks including grammar correction, spelling checks, and minor phrasing improvements to enhance clarity and concision.

## B    Datasets

Figure 6 presents representative samples from each class across the three datasets employed in our study: ISIC, DeepDRiD, and PANDA. We display one exemplar image per class, organized with rows corresponding to individual datasets and columns representing distinct classes. This visualization facilitates direct comparison of class-specific visual characteristics. The corresponding text prompts utilized for training CLIP models are detailed in Table 5, including auxiliary class prompts for each dataset.

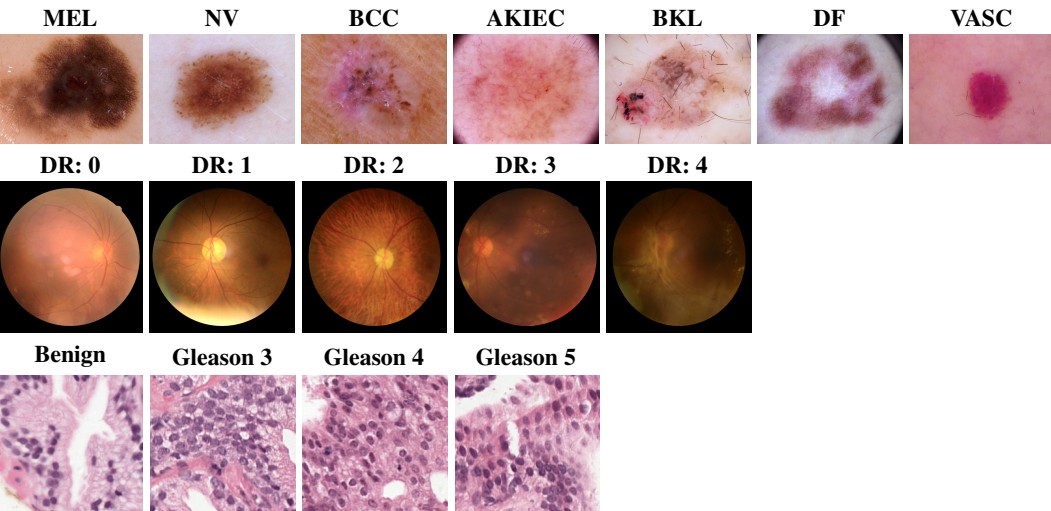

Figure 6: Representative images from the datasets used in this work. Rows correspond to ISIC, DeepDRiD, and PANDA datasets (top to bottom) and columns to class labels.

Table 5: Text prompts for each class in the datasets. Prompts highlighted in gray are auxiliary class prompts.

| Dataset | Text Prompts |
|---|---|
| ISIC | A dermoscopic image showing melanoma. |
| | A dermoscopic image showing nevus. |
| | A dermoscopic image showing basal cell carcinoma. |
| | A dermoscopic image showing actinic keratosis/intraepithelial carcinoma. |
| | A dermoscopic image showing benign keratosis. |
| | A dermoscopic image showing dermatofibroma. |
| | A dermoscopic image showing vascular lesion. |
| | A dermoscopic image showing other lesions. |
| DeepDRiD | A fundus image showing no evidence of diabetic retinopathy. |
| | A fundus image exhibiting mild diabetic retinopathy. |
| | A fundus image exhibiting moderate diabetic retinopathy. |
| | A fundus image exhibiting severe diabetic retinopathy. |
| | A fundus image exhibiting proliferative diabetic retinopathy. |
| | A fundus image showing other retinal conditions. |
| PANDA | A histology image showing benign glandular epithelium. |
| | A histology image showing Gleason pattern 3 adenocarcinoma. |
| | A histology image showing Gleason pattern 4 adenocarcinoma. |
| | A histology image showing Gleason pattern 5 adenocarcinoma. |
| | A histology image showing other conditions. |

## C  ADDITIONAL ANALYSIS OF ENTROPY TRAJECTORIES

In this section, we visualize entropy trajectories for more representative categories across the three datasets. For ISIC and DeepDRiD, we additionally show trajectories for other classes, comparing correctly labeled samples with mislabeled ones. As shown in Figure 7, we plot entropy trajectories for melanoma cases from the ISIC dataset, grade 0 diabetic retinopathy images from the DeepDRiD dataset, benign epithelium samples from the PANDA dataset, and Gleason 5 cancerous epithelium images from the PANDA dataset. Across all examined cases, we consistently observe the regularity described in Section 2.3.

## D  EXTENDED EVALUATION OF SEI FOR MISLABELED SAMPLE DETECTION

We present additional empirical evidence demonstrating the effectiveness of SEI. Figures 8, 9, and 10 illustrate the discriminative power of SEI in separating different sample types within the ISIC, DeepDRiD, and PANDA datasets, respectively. The results consistently validate the theoretical framework outlined in Section 2.4.2: samples with correct labels that are easily classified exhibit large positive SEI values, challenging but correctly labeled samples demonstrate moderate SEI values, while mislabeled samples consistently display strongly negative SEI values.

## E  THEORETICAL PROPERTIES OF SIGNED ENTROPY

In this section, we provide a short theoretical analysis of the proposed signed entropy (Eq. 2). Recall that for $(\boldsymbol{x}, y) \in \mathcal{D}_{\texttt{train}}$ with posterior $\boldsymbol{p}(\boldsymbol{x})$, we define

$$\mathcal{H}(\boldsymbol{p}(\boldsymbol{x}), y) = (-1)^{\mathbb{1}[y=\arg\max_k p_k(\boldsymbol{x})]} \sum_{k=1}^{K} p_k(\boldsymbol{x}) \log p_k(\boldsymbol{x}) \,.$$

### E.1  RELATION TO SHANNON ENTROPY

**Proposition 1 (Reduction to Shannon Entropy)** *If $y = \arg\max_k p_k(\boldsymbol{x})$, then*

$$\mathcal{H}(\boldsymbol{p}(\boldsymbol{x}), y) = H(\boldsymbol{p}(\boldsymbol{x})) \,,$$

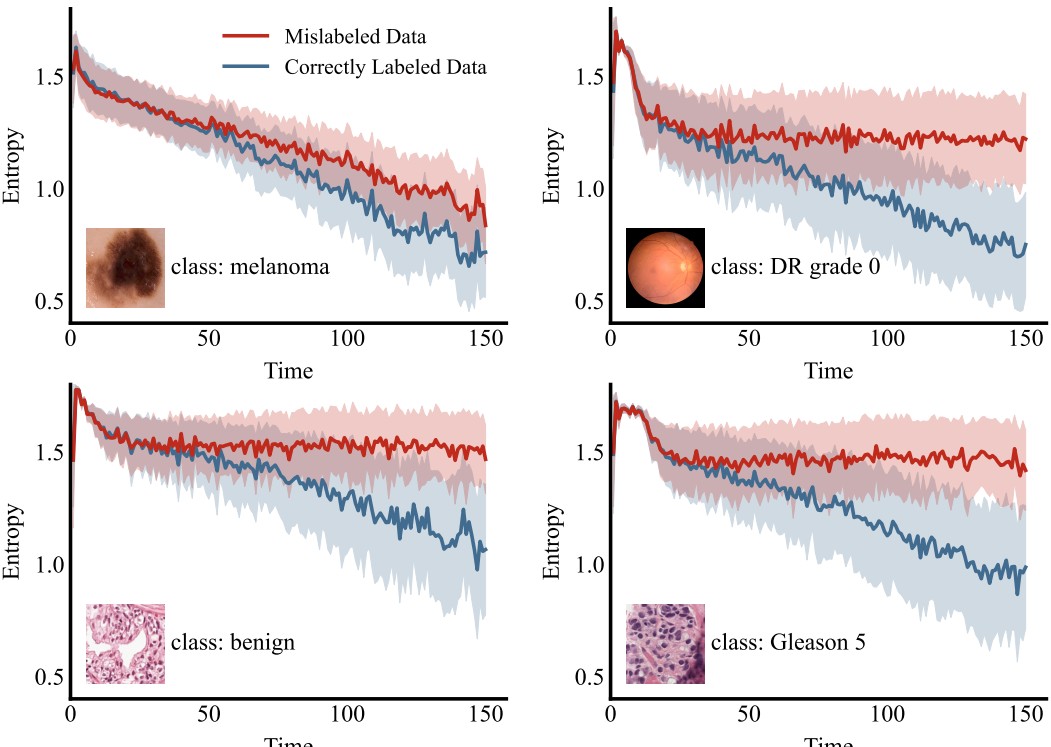

Figure 7: Training dynamics of prediction entropy for correctly labeled versus mislabeled samples. Top-left: melanoma images from the ISIC dataset. Top-right: grade 0 diabetic retinopathy (DR) images from the DeepDRiD dataset. Bottom-left: benign glandular epithelium images from the PANDA dataset. Bottom-right: Gleason 5 cancerous epithelium images from the PANDA dataset.

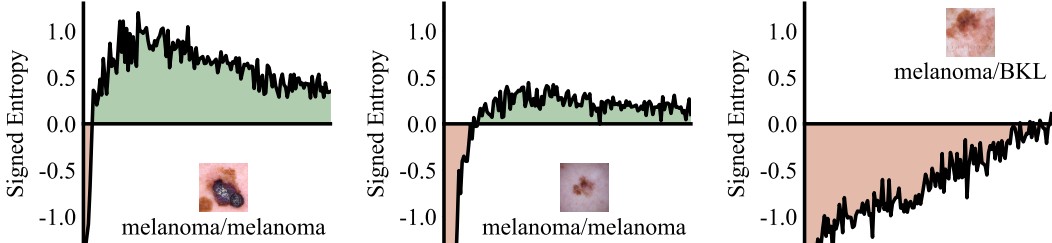

Figure 8: Illustration of SEI using melanoma images from the ISIC dataset. The plots show signed entropy curves across training epochs for easy clean (left), hard clean (middle), and mislabeled (right) samples. Each curve is averaged over 200 samples, and the signed area under the curve represents the SEI. Correctly labeled samples consistently exhibit larger SEIs than mislabeled ones.

where $H(\boldsymbol{p}) = -\sum_k p_k \log p_k$ is Shannon's entropy.

**Proof 1** *By definition, the sign exponent equals $(-1)^1 = -1$ when the prediction agrees with $y$, yielding the standard Shannon entropy.*

### E.2  CONCAVITY AND SIGN SYMMETRY

**Proposition 2 (Concavity up to Sign)** *Let $\mathcal{P}$ denote the probability simplex in $\mathbb{R}^K$. For a fixed label $y$, the signed entropy $\mathcal{H}(\cdot, y)$ is concave on $\mathcal{P}$ if $y$ matches the prediction, and is convex on $\mathcal{P}$ if $y$ disagrees with the prediction.*

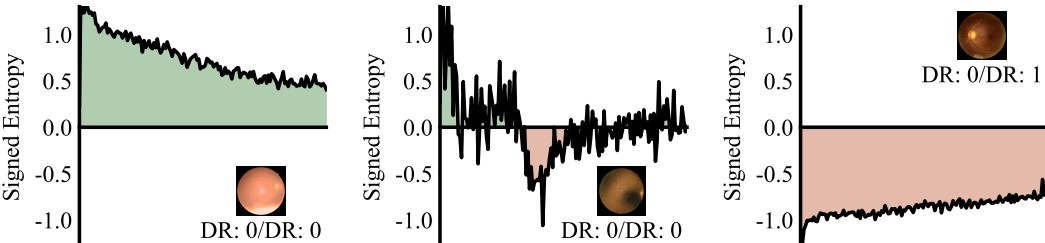

Figure 9: Illustration of SEI using Grade 0 diabetic retinopathy (DR) images from the DeepDRiD dataset.

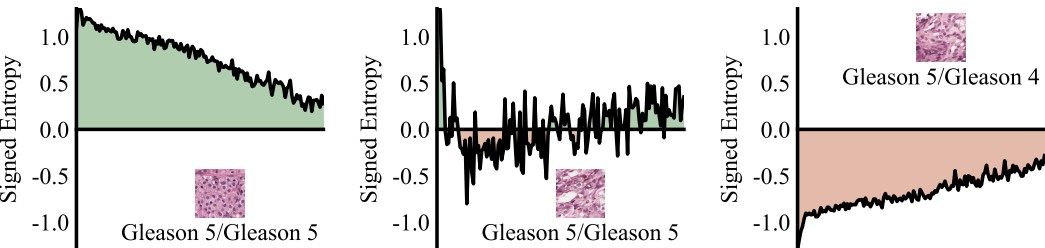

Figure 10: Illustration of SEI using Gleason 5 images from the PANDA dataset.

**Proof 2** *The Shannon entropy $H(\boldsymbol{p})$ is strictly concave on $\mathcal{P}$ (classical result). Multiplying by $-1$ flips concavity to convexity. Since the sign of $\mathcal{H}$ depends only on alignment, the stated property follows.*

### E.3 IMPLICATION FOR SEI

These properties imply that SEI (Eq. 3) can be interpreted as a signed, temporally averaged measure of prediction uncertainty. Its sign encodes long-term label alignment, while its magnitude captures how confidently the model reaches this alignment (or misalignment). This dual role is what enables SEI to separate mislabeled data from both easy and hard clean samples.

## F ADDITIONAL ABLATION RESULTS

### F.1 VISUAL EVIDENCE FOR THE SIGNED TERM

We further visualize score distributions for clean and mislabeled samples under both formulations (see Figure 11). The unsigned variant exhibits heavily overlapping positive-only distributions, making separation difficult. In contrast, SEI introduces a clear negative tail for mislabeled samples, creating a bimodal structure with reduced overlap and more distinguishable groups.

### F.2 FULL-TRAJECTORY INTEGRATION VS. WINDOWED INTEGRALS

Table 6 shows that both windowed variants (SEI@Early and SEI@Late) perform worse than the full-trajectory SEI. Restricting to either an early or late window discards complementary cues present in the other phase.

## G ADDITIONAL RESULTS ON DOWNSTREAM IMAGE CLASSIFICATION

Beyond F1 score, we also report accuracy and AUC for SCE, M-correction, DivideMix, and ProMix, both with and without SEI, under the same protocol as Section 4.3.4. Results are presented in Figure 12 and Figure 13.

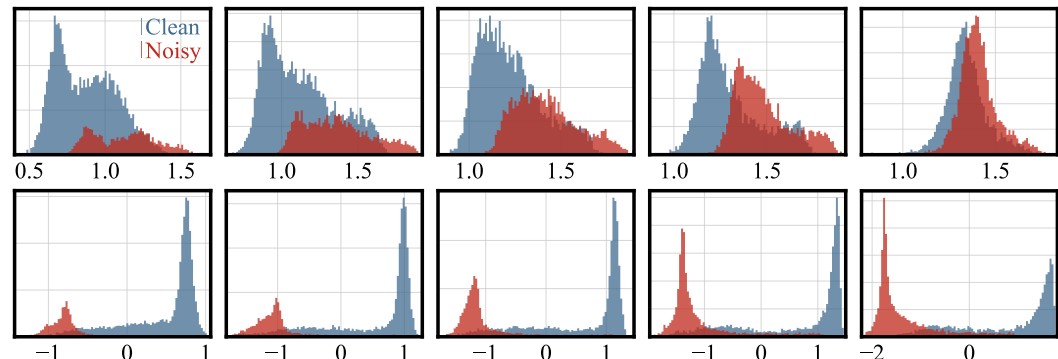

Figure 11: Score distributions for clean and noisy samples on the PANDA dataset. We compare our proposed SEI statistic (bottom row) against the unsigned Shannon entropy integral baseline (top row) for noise rates $\eta \in \{0.1, 0.2, 0.3, 0.4, 0.5\}$. Each column corresponds to increasing noise levels from left to right. The SEI statistic demonstrates better separation between clean and noisy sample distributions at all noise levels.

Table 6: Evaluation of windowed integrals—SEI@Early (epochs 1–75) and SEI@Late (epochs 76–150)—compared to the full SEI.

| | ISIC | | | | | DeepDRiD | | | | | PANDA | | | | |
|---|---|---|---|---|---|---|---|---|---|---|---|---|---|---|---|
| | 0.1 | 0.2 | 0.3 | 0.4 | 0.5 | 0.1 | 0.2 | 0.3 | 0.4 | 0.5 | 0.1 | 0.2 | 0.3 | 0.4 | 0.5 |
| SE@Early | 44.88 | 56.15 | 63.50 | 69.88 | 73.80 | 36.05 | 42.69 | 54.30 | 60.63 | 66.48 | 68.01 | 70.89 | 75.05 | 76.36 | 76.11 |
| SE@Late | 37.98 | 51.23 | 62.57 | 65.77 | 69.56 | 41.11 | 47.08 | 58.81 | 63.96 | 70.81 | 70.46 | 73.50 | 77.10 | 77.91 | 78.66 |
| SEI | **47.75** | **59.35** | **67.25** | **74.98** | **79.91** | **46.59** | **52.65** | **62.84** | **68.35** | **73.04** | **73.17** | **78.21** | **81.86** | **81.96** | **81.85** |

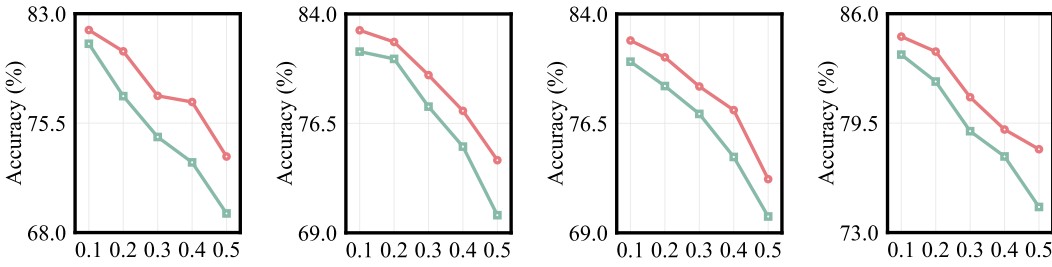

Figure 12: Accuracy comparison under confusion-calibrated noise between baseline noisy label learning methods (green) and their SEI-enhanced variants (red). From left to right: SCE, M-correction, DivideMix, and ProMix.

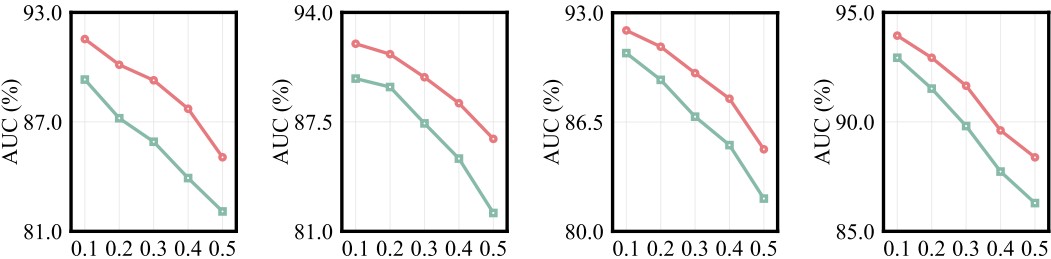

Figure 13: AUC comparison under confusion-calibrated noise between baseline noisy label learning methods (green) and their SEI-enhanced variants (red). From left to right: SCE, M-correction, DivideMix, and ProMix.

In addition, we provide comprehensive results (F1 score, accuracy, and AUC) under symmetric noise, which are reported in Figures 14, 15, and 16.

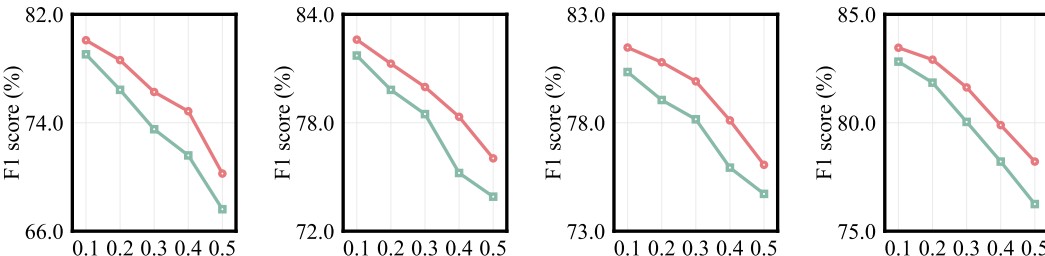

Figure 14: F1 score comparison under symmetric noise between baseline noisy label learning methods (green) and their SEI-enhanced variants (red). From left to right: SCE, M-correction, DivideMix, and ProMix.

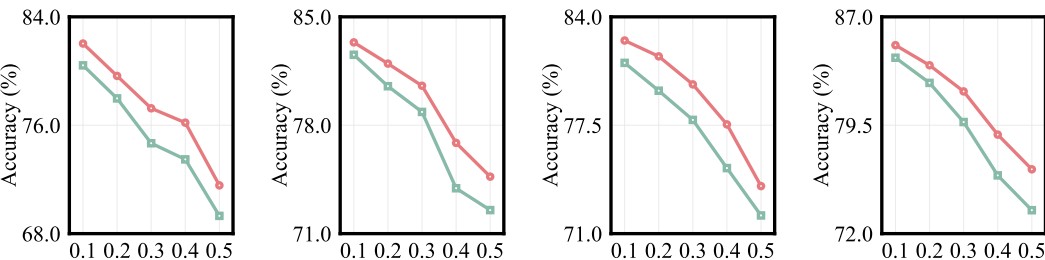

Figure 15: Accuracy comparison under symmetric noise between baseline noisy label learning methods (green) and their SEI-enhanced variants (red). From left to right: SCE, M-correction, DivideMix, and ProMix.

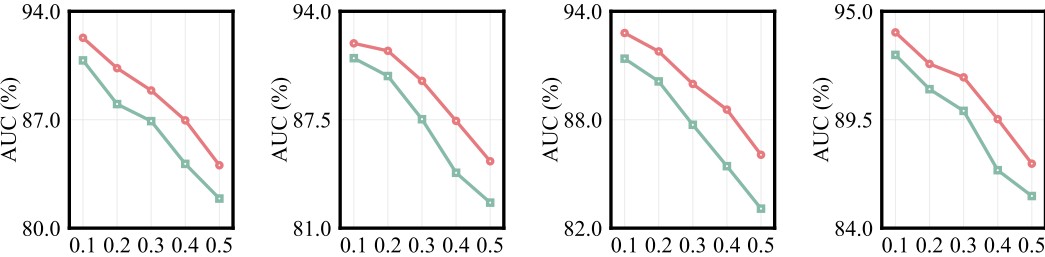

Figure 16: AUC comparison under symmetric noise between baseline noisy label learning methods (green) and their SEI-enhanced variants (red). From left to right: SCE, M-correction, DivideMix, and ProMix.

## H  SENSITIVITY TO TRAINING CONFIGURATION

We conducted a series of hyperparameter sensitivity experiments on the PANDA dataset under the confusion-calibrated 50% noise setting to assess how SEI behaves under different training configurations.

**Training duration and early stopping.**   We first varied the number of training epochs while keeping all other hyperparameters fixed. As shown in Table 7, the small standard deviation (0.42%) indicates that SEI is insensitive to moderate changes in training duration as long as the model reaches convergence. In contrast, early stopping at epoch 75 leads to a noticeable drop (Table 6), confirming that SEI assumes a reasonably converged model rather than an undertrained one.

**Learning rate.**   We investigate the sensitivity of SEI to the choice of initial learning rate by sweeping it from 1e-4 to 1e-2. As shown in Table 8, we observe that the performance of SEI remains

Table 7: Sensitivity of SEI to training duration on PANDA (confusion-calibrated noise, $\eta = 0.5$). The default setting is 150 epochs.

| Epochs | 100 | 150 | 200 | mean | std |
|---|---|---|---|---|---|
| F1 (%) | 81.34 | 81.85 | 81.02 | 81.40 | 0.42 |

very stable within the range typically used for training CLIP and standard classifiers (5e-4–1e-3), and only starts to degrade when the learning rate is excessively large (5e-3 or 1e-2). This trend mirrors standard classification training, where an appropriate learning rate is needed for effective learning of the base model itself.

Table 8: Sensitivity of SEI to learning rate on PANDA (confusion-calibrated noise, $\eta = 0.5$). The default learning rate is 1e-3.

| Learning rate | 1e-4 | 5e-4 | 1e-3 | 5e-3 | 1e-2 |
|---|---|---|---|---|---|
| F1 (%) | 80.06 | 81.50 | 81.85 | 78.86 | 77.00 |

**Data augmentation, label smoothing, and weight decay.** We also examined several common regularization choices. For data augmentation, as shown in Table 9, "Strong aug." adds random affine transforms and random erasing. SEI remains robust under both weak and strong augmentations, with a slight gain under stronger augmentation. MixUp—whose heavy mixing disrupts label–prediction alignment—slightly reduces performance. For label smoothing, as shown in Table 10, moderate label smoothing slightly degrades performance, and stronger smoothing further weakens entropy signals, as expected. As shown in Tabel 11, weight decay within [5e-5, 5e-4] has only mild impact, with slightly higher F1 at the upper end.

Table 9: Effect of data augmentation on SEI (PANDA, confusion-calibrated noise, $\eta = 0.5$). The default setting is weak augmentation.

| Augmentation setting | Strong aug. | Weak aug. | MixUp |
|---|---|---|---|
| **F1 (%)** | 82.38 | 81.85 | 80.88 |

Table 10: Effect of label smoothing on SEI (PANDA, confusion-calibrated noise, $\eta = 0.5$). The default setting is 0.0.

| Label smoothing | 0.0 | 0.1 | 0.2 |
|---|---|---|---|
| **F1 (%)** | 81.85 | 81.45 | 80.36 |

Table 11: Effect of weight decay on SEI (PANDA, confusion-calibrated noise, $\eta = 0.5$). The default setting is $1e - 4$.

| Weight decay | $5e - 5$ | $1e - 4$ | $5e - 4$ |
|---|---|---|---|
| **F1 (%)** | 81.36 | 81.85 | 82.21 |

**Temporal integration window.** We compare the full-trajectory SEI with variants that integrate signed entropy over fixed 50-epoch windows with a stride of 20. As reported in Table 12, all window-based variants perform significantly worse than full-trajectory SEI, consistent with the

Table 12: SEI with different 50-epoch integration windows (PANDA, confusion-calibrated noise, $\eta = 0.5$).

| Window (epochs) | [0, 49] | [20, 69] | [40, 89] | [60, 109] | [80, 129] | [100, 149] |
|---|---|---|---|---|---|---|
| F1 (%) | 75.34 | 50.78 | 49.92 | 51.03 | 54.09 | 56.81 |

SEI@Early/SEI@Late results in Table 6. This confirms that SEI benefits from integrating the entire training trajectory rather than relying on a narrow early or late slice.

In summary, SEI is robust under standard training-to-convergence settings and typical hyperparameter choices. The ranking is stable with respect to reasonable training lengths, LR schedules, and regularization. Our recommendation is to use the full training trajectory; extreme early stopping or highly atypical hyperparameters are not ideal conditions for SEI.

## I    SENSITIVITY TO THE AUXILIARY-CLASS SAMPLING RATIO

To evaluate the robustness of the auxiliary-class–based threshold, we performed a sensitivity analysis on the PANDA dataset with 50% confusion-calibrated noise by varying the sampling ratio of auxiliary-class samples. In our default setting, the number of auxiliary samples is N/(K+1). We scaled this number using factors of 0.5×, 0.75×, 1.0×, 1.5×, and 2.0×. As shown in Tabel 13, the F1 scores remain within a tight range, with a mean of 81.87% and a standard deviation of only 0.72%, even when the number of auxiliary samples varies by a factor of four. This indicates that SEI is not sensitive to the exact sampling ratio: as long as a reasonable number of auxiliary samples is used, the estimated mean SEI for the auxiliary class remains stable. It is also worth noting that although the sampling ratio is manually specified, the threshold itself is entirely data-driven and learned adaptively.

Table 13: Sensitivity of SEI to the auxiliary-class sampling ratio on PANDA (confusion-calibrated noise, $\eta = 0.5$). The ratio is expressed as a multiplier of the default $N/(K + 1)$ setting. The default setting is 1.0×.

| Ratio factor | 0.5× | 0.75× | 1.0× | 1.5× | 2.0× |
|---|---|---|---|---|---|
| F1 (%) | 81.94 | 83.01 | 81.85 | 81.41 | 81.12 |

## J    GENERALIZATION OF SEI TO NATURAL IMAGES WITH REAL-WORLD NOISE

To assess the generalization ability of SEI beyond the medical domain and to evaluate its behavior under real-world human annotation noise, we conducted additional experiments on the CIFAR-100N (Wei et al., 2022), a natural-image dataset with real human annotation noise. CIFAR-100N is a noisy-label variant of CIFAR-100 where each training image is re-annotated by human annotators, while the original CIFAR-100 labels are kept as clean ground truth.

On CIFAR-100N, we compare SEI against the same set of noisy label detection baselines used in the main paper. As shown in Table 14, SEI again achieves the best F1 score for mislabeled data detection, outperforming the second-best method by 2.2%. These results indicate that SEI is not domain-specific. The signed entropy dynamics remain effective on a standard natural image dataset, and SEI robustly handles real human annotation noise—not only synthetic symmetric or confusion-calibrated noise.

## K    INTER-OBSE VARIABILITY VS. LABEL NOISE

Table 14: Mislabeled sample detection on CIFAR-100N. We report F1 scores (%) for identifying mislabeled samples. The best result is highlighted in **bold**.

| Method | INCV | BMM | GMM | AUM | CORES | CL | SIMIFEAT | DeFT | ReCoV | LEMoN | Ours |
|--------|------|-----|-----|-----|-------|-----|----------|------|-------|-------|------|
| F1 (%) | 59.77 | 63.55 | 63.83 | 74.54 | 38.52 | 67.64 | 79.21 | 75.03 | 67.59 | 78.40 | **81.41** |

Inter-observer variability and label noise are related but not identical. Variability across annotators often reflects inherent uncertainty, whereas noisy label detection assumes a single hard label per sample and seeks to identify cases where that label is incorrect. Learning with uncertain or probabilistic labels—such as modeling annotator distributions or ambiguity—is an important but distinct problem setting (Kohl et al., 2018) and is not the focus of this work.

Our work operates strictly under the standard hard-label noise detection setting, where each training sample is associated with one label, and the goal is to detect mislabeled instances under this assumption. SEI is therefore designed and evaluated within this framework.

## L  SEI under Extreme Noise Rates

In clinical practice, datasets with more than 50% label disagreement are typically considered unreliable for training. Accordingly, our main experiments focus on noise rates up to 0.5. To further assess robustness, we additionally evaluate SEI on the PANDA dataset under more extreme noise levels $\eta \in \{0.6, 0.7, 0.8\}$, for both confusion-calibrated noise and symmetric noise. The F1 scores for mislabeled-data detection are summarized in Table 15.

Table 15: SEI performance on PANDA under extreme noise rates. We report F1 scores (%) for mislabeled data detection.

| Noise rate | 0.6 | 0.7 | 0.8 |
|------------|-----|-----|-----|
| Confusion-calibrated | 81.07 | 79.93 | 81.83 |
| Symmetric | 81.66 | 80.95 | 80.51 |

Even at very high noise rates (60—80%), SEI remains stable around 80% F1, without a significant performance collapse. Performance does not degrade monotonically; instead, it fluctuates slightly within a narrow band, suggesting that SEI can still capture useful training-dynamics signals even when a large fraction of labels is corrupted.

## M  Per-Class Analysis of Noisy-Label Detection

We perform a per-class analysis of noisy label detection performance on PANDA with 50% confusion-calibrated noise. For each class, we report the class-wise false positive rate (FPR) and per-class F1 score as shown in Table 16.

Table 16: Per-class false positive rate and F1 for noisy-label detection on PANDA (confusion-calibrated noise, $\eta = 0.5$).

| Class | Per-class FPR (%) | Per-class F1 (%) |
|-------|-------------------|------------------|
| Benign epithelium | 13.66 | 85.47 |
| Gleason 3 | 22.87 | 80.38 |
| Gleason 4 | 25.87 | 80.14 |
| Gleason 5 | 21.63 | 82.91 |

These results show that SEI does not systematically over-filter any particular class. Notably, benign epithelium—the smallest class—has the lowest false positive rate (13.66%), suggesting that minority

patterns are not disproportionately removed. Overall, SEI achieves consistently strong detection quality across all classes.

This outcome aligns with the design of SEI: by integrating signed entropy over the entire training trajectory, SEI naturally separates hard-but-correct samples from mislabeled ones. Hard, correctly labeled samples may have high entropy early in training but eventually align with their labels and accumulate positive signed contributions. In contrast, mislabeled samples remain misaligned for most of training and accumulate negative contributions. This reduces the risk of misclassifying intrinsically hard or minority-subtype samples as noisy.

