# OpenReview forum: "Entropy Never Lies: Signed Entropy Integral Unmasks Mislabeled Data"
_ICLR.cc/2026/Conference — Submitted to ICLR 2026_

### Official Review · Reviewer_kas8 · 2025-10-30

**Soundness:** 2
**Presentation:** 3
**Contribution:** 2
**Rating:** 2
**Confidence:** 3

**Summary:**

The paper addresses the problem of distinguishing mislabelled (noisy) samples from valuable clean samples that are difficult to classify. Motivated by medical data, it demonstrates that both mislabelled and clean samples exhibit high prediction entropy during training, which makes them difficult to distinguish. However, it is also observed that they have different training dynamics. To make use of this information, the paper proposes defining a 'signed entropy' statistic for self-supervision by introducing a sign function on top of the Shannon entropy. Additionally, the signed entropy integral is introduced to leverage the training dynamics across the entire training trajectory (rather than just one iteration). Finally, a data-driven threshold is used to separate clean samples from mislabelled ones. This approach was evaluated using three noisy label learning medical datasets and showed promising results.

**Strengths:**

The idea is simple yet effective. It demonstrates its effectiveness with noisy and difficult-to-clean labels.

The paper is very well written and easy to follow. The proposed method is intuitively clear and well motivated.

The ablation studies effectively support the claim that the proposed sign is necessary for the entropy function.

In theory, the proposed method could be applied to any type of data, not just medical data.

**Weaknesses:**

The paper is based only on medical datasets. There are standard benchmarks for noisy label learning to demonstrate generalisation, such as Clothing1M and WebVision, as well as small-scale datasets.

The paper is based purely on synthetic noisy scenarios. This is another clear limitation that needs to be addressed using real-world noise.

It would be helpful to make comparisons with entropy-based loss variations or temporal entropy integration. There are many methods for modifying standard entropy loss. For example, 'Learning from Training Dynamics: Identifying Mislabelled Data Beyond Manually Designed Features' (AAAI 2023) and 'Efficient Adaptive Label Refinement for Label Noise Learning' (Neurocomputing 2025).

The method can be computationally expensive due to the integration over training time. Presenting large-scale datasets would demonstrate this in practice.

The presented threshold is automatic, but is based purely on heuristics. This implies the need for particular tuning for different types of data. This is another reason for evaluation on standard benchmarks.

**Questions:**

Why was the real-world noise experiment skipped?

Also, why are standard noisy label learning benchmarks missing?

---

> ### Author Response · Authors · 2025-11-27
>
> > The paper is based only on medical datasets. There are standard benchmarks for noisy label learning to demonstrate generalisation, such as Clothing1M and WebVision, as well as small-scale datasets.
> The paper is based purely on synthetic noisy scenarios. This is another clear limitation that needs to be addressed using real-world noise.
> Why was the real-world noise experiment skipped?
>
> We appreciate the reviewer’s comments and thanks for pointing out these datasets. With full respect, we would like to clarify that Clothing1M and WebVision are commonly used for learning with noisy labels, but they are not suitable for the mislabeled data detection task studied in this work. This is because they either lack clean ground-truth labels for all samples or only provide small clean subsets. Reliable ground truth is required to quantitatively evaluate mislabeled sample identification.
>
> To address the reviewer’s concern and demonstrate both SEI’s generalization beyond medical imaging and its effectiveness under real-world human annotation noise, we have added new experiments on the CIFAR-100N [1] dataset (Appendix J, “Generalization of SEI to Natural Images with Real-World Noise”), with the added text highlighted in blue in the revised manuscript.
>
> On CIFAR-100N, we compare SEI against the same set of noisy label detection baselines used in the main paper. As shown in Table 14 of the revised manuscript, SEI again achieves the best F1 score for mislabeled data detection, outperforming the second-best method by 2.2%. The full comparison is:
>
> | **Method**| **INCV** | **BMM** | **GMM** | **AUM** | **CORES** | **CL** | **SIMIFEAT** | **DEFT** | **ReCoV** | **LEMoN** | **Ours** |
> |:--------|:--------:|:--------:|:--------:|:--------:|:----------:|:------:|:------------:|:--------:|:---------:|:---------:|:--------:|
> | F1 (%) | 59.77 | 63.55 | 63.83 | 74.54 | 38.52 | 67.64 | 79.21 | 75.03 | 67.59 | 78.40 | **81.41** |
>
> These results indicate that SEI is not domain-specific. The signed entropy dynamics remain effective on a standard natural image dataset, and SEI robustly handles real human annotation noise—not only synthetic symmetric or confusion-calibrated noise.
>
> [1] Jiaheng Wei, Zhaowei Zhu, Hao Cheng, Tongliang Liu, Gang Niu, and Yang Liu. Learning with noisy labels revisited: A study using real-world human annotations. In ICLR, 2022.

---

> ### Author Response · Authors · 2025-11-27
>
> > It would be helpful to make comparisons with entropy-based loss variations or temporal entropy integration. There are many methods for modifying standard entropy loss. For example, ‘Learning from Training Dynamics: Identifying Mislabelled Data Beyond Manually Designed Features’ (AAAI 2023) and ‘Efficient Adaptive Label Refinement for Label Noise Learning’ (Neurocomputing 2025).
>
> We thank the reviewer for pointing out these references.
>
> **AAAI 2023 (Learning from Training Dynamics):** This method requires supervised information during training, i.e., knowledge of which samples are mislabeled versus correctly labeled. In contrast, our method—as well as all competing methods evaluated in our experiments—is fully unsupervised, without any access to such labels. Comparing a supervised method with unsupervised methods would therefore be unfair and not aligned with our task setting.
>
> **Neurocomputing 2025 (Efficient Adaptive Label Refinement):** This method focuses on learning with noisy labels, rather than mislabeled data identification. Consequently, it is not directly comparable to our approach.

---

> ### Author Response · Authors · 2025-11-27
>
> > The method can be computationally expensive due to the integration over training time. Presenting large-scale datasets would demonstrate this in practice.
>
> We respectfully note that, despite using information “over training time”, SEI is in fact computationally very lightweight. Our design goal was specifically to keep the method simple and to avoid altering the existing training paradigm of the base model.
>
> Concretely:
>
> 1. SEI does not require any extra forward or backward passes beyond standard training.
> 2. For each sample, we only maintain a running sum of signed entropy—i.e., a single scalar updated per sample per epoch. This is implemented online without storing per-epoch predictions, so the extra memory cost is just O(N) scalars.
> 3. The signed entropy is computed from the same class probabilities already used for the cross-entropy loss, adding only a few elementwise operations per sample per batch.
>
> Empirically, on our three medical datasets, wall-clock training time with SEI is nearly identical to standard training, with only a small bookkeeping overhead. For example, on PANDA, the average epoch time increases from 127.6 seconds (standard training) to 135.9 seconds (with SEI), corresponding to only about 6.5% overhead per epoch. In contrast, many existing methods introduce substantial extra cost (e.g., multi-stage pipelines, co-training multiple networks, or repeated re-labeling), whereas SEI remains a single-pass, plug-and-play statistic with negligible additional computation.

---

> ### Author Response · Authors · 2025-11-27
>
> > The presented threshold is automatic, but is based purely on heuristics. This implies the need for particular tuning for diferent types of data. This is another reason for evaluation on standard benchmarks.
>
> We apologize for any lack of clarity. With full respect, our thresholding scheme is manually defined in form but not manually tuned: the threshold value is data-driven and adaptively learned from SEI statistics. Across all experiments in the paper—including all three datasets, all noise types, and all noise rates—we use the same auxiliary-class design and the same sampling ratio N/(K+1), without any dataset-specific tuning.
>
> To further address the reviewer’s concern, we conducted an ablation on the auxiliary sampling ratio on PANDA with 50% noise, scaling it around the default N/(K+1):
>
> |  | **0.5×** | **0.75×** | **1.0×** | **1.5×** | **2.0×** | **mean** | **std** |
> |:--------|:-------:|:--------:|:-------:|:-------:|:-------:|:------:|:------:|
> | F1 (%)   | 81.94 | 83.01 | 81.85 | 81.41 | 81.12 | 81.87 | 0.72 |
>
>
> Performance remains within a very narrow range, with a mean of 81.87% and a standard deviation of 0.72%, demonstrating that SEI is insensitive to the exact sampling ratio and that the thresholding mechanism is robust rather than finely tuned for each dataset. We have clarified in Appendix I of the revised manuscript that the threshold is learned in a data-driven manner and that the same ratio is used across all datasets and noise settings.

---

> ### Author Response · Authors · 2025-11-27
>
> > Also, why are standard noisy label learning benchmarks missing?
>
> We thank the reviewer for raising the question.
>
> 1. We would like to emphasize that our focus on medical datasets is deliberate. This work is conducted as part of a project specifically aimed at improving label reliability in medical imaging, and our goal is to address noisy label detection in this setting. Medical labels are costly and specialized, often requiring board-certified clinicians, multi-reader consensus, or histopathology confirmation. As a result, large-scale relabeling or exhaustive manual cleaning is rarely feasible, and even modest systematic noise can carry direct clinical implications. In this context, a method that reliably flags suspicious labels for expert review is itself of significant practical value.
>
>
> 2. To further address the reviewer’s concern regarding evaluation on standard benchmarks, we have added new experiments on the CIFAR-100N dataset in Appendix J (as noted in a previous response).

---

### Official Review · Reviewer_GH2B · 2025-11-01

**Soundness:** 3
**Presentation:** 4
**Contribution:** 2
**Rating:** 4
**Confidence:** 4

**Summary:**

This paper introduces SEI, a signed-entropy–based training-dynamics metric to detect mislabeled samples, and reports state-of-the-art results on three medical imaging datasets. The core problem tackled is that overparameterized networks tend to memorize noisy labels, degrading performance; detecting and filtering mislabeled data during training is therefore critical. The key idea is that correctly labeled samples show a consistent drop in prediction entropy across epochs, whereas mislabeled samples maintain relatively high entropy; SEI integrates (signed) entropy over time to capture both magnitude and trend, with the sign reflecting label–prediction consistency. The implementation is simple and the method plugs into standard training loops. The claimed conclusion is that SEI is a simple, efficient, and broadly applicable metric that achieves SOTA mislabeled-data identification without architectural changes or complex training procedures.

**Strengths:**

- The paper is clearly written and has a well-structured presentation. Figures and tables are clean and directly support the claims; the narrative is easy to follow.
- The empirical section is strong within the chosen three datasets: ablations and compatibility checks against other SOTA training pipelines.
- The method is simple and practical. No architectural changes needed; it integrates into existing training workflows with low overhead. This makes it work well with CLIP fine-tuning and, in principle, with standard classifiers.
- The use of entropy trajectories makes the decision process intuitive and easy to reason about. This gives the approach an interpretability angle.

**Weaknesses:**

- Limited novelty relative to prior entropy-based signals. The signed and temporal integration is a neat spin, but the conceptual jump may be incremental given prior work on entropy/uncertainty and training-dynamics signals.
- The paper has a narrow domain scope. All results are on medical imaging; it’s unclear whether the approach generalizes to broader CV/ML settings. This also challenges the impact of the SOTA results and brings forward a question about how competitive is SEI when evaluated in standard benchmarks for label noise.
- Similarly to the last point, the evaluation relies too much on synthetic noise. If the evaluation relies mainly on synthetic label noise, the conclusions may not carry over to real-world noisy datasets; this gap is well-documented in the literature.
- SEI depends on training dynamics; thresholds and rankings might vary with epoch budget, LR schedules, data augmentation, label smoothing, or heavy regularization. This could be addressed with further discussion or ablation studies exploring the different design decisions made.

**Questions:**

- Beyond CLIP, how does SEI perform with standard CNNs/ViTs trained from scratch or with supervised pretraining? Did the authors observe any meaningful differences in how different architectures present different entropy values/tendencies in noisy datasets?
- How sensitive is SEI ranking to training length (early vs late epochs), LR schedules, strong augmentation, label smoothing, and weight decay? Is there a recommended epoch window for a stable ranking?
- Samples that are intrinsically hard or minority-pattern but correctly labeled may retain higher entropy and risk being flagged as noisy (class imbalance/long-tail scenarios). Do you have analyses showing that SEI does not systematically filter rare but correctly labeled patterns (e.g., minority subtypes)? Any per-class or per-subpopulation error analysis?
- In which regimes does SEI struggle? Have you explored more extreme noise rates or class imbalances?

---

> ### Author Response · Authors · 2025-11-27
>
> > Limited novelty relative to prior entropy-based signals. The signed and temporal integration is a neat spin, but the conceptual jump may be incremental given prior work on entropy/uncertainty and training-dynamics signals.
>
> We appreciate the reviewer’s observation that our work builds on a long line of research using entropy and training dynamics. While entropy itself is not new, our contribution lies in how we reformulate and operationalize it for mislabeled-sample detection.
>
> SEI combines three key ingredients in a way that, to the best of our knowledge, is distinct from prior work:
>
> **1. Signed entropy:** Instead of using raw entropy or loss alone, we introduce a label-consistency–aware extension of Shannon entropy that flips sign when the model prediction disagrees with the given label. This produces a directional quantity encoding “agreement vs. disagreement,” rather than just uncertainty.
>
> **2. Full-trajectory temporal integration:** Rather than relying on single-epoch snapshots, we integrate the signed entropy over the entire training course into a single scalar SEI. Ablation studies (Tables 3 and 6) show that this simple integration substantially improves separation between hard-but-correct and mislabeled samples compared with unsigned entropy or single-epoch/windowed variants.
>
> **3. Adaptive thresholding mechanism:** SEI is tied to a data-driven calibration procedure via an auxiliary class, providing an automatic decision threshold without per-dataset tuning, as supported by our ratio-sensitivity study.
>
> Compared with prior training dynamics-based methods—which often require multi-stage pipelines, additional networks, or manually designed trajectory features—SEI is architecture-agnostic and plug-and-play in practice.

---

> ### Author Response · Authors · 2025-11-27
>
> > The paper has a narrow domain scope. All results are on medical imaging; it’s unclear whether the approach generalizes to broader CV/ML settings. This also challenges the impact of the SOTA results and brings forward a question about how competitive is SEI when evaluated in standard benchmarks for label noise.
> Similarly to the last point, the evaluation relies too much on synthetic noise. If the evaluation relies mainly on synthetic label noise, the conclusions may not carry over to real-world noisy datasets; this gap is well-documented in the literature.
>
> We thank the reviewer for this valuable comment.
>
> 1. We would like to emphasize that our focus on medical datasets is deliberate. This work is conducted as part of a project specifically aimed at improving label reliability in medical imaging, and our goal is to address noisy label detection in this setting. Medical labels are costly and specialized, often requiring board-certified clinicians, multi-reader consensus, or histopathology confirmation. As a result, large-scale relabeling or exhaustive manual cleaning is rarely feasible, and even modest systematic noise can carry direct clinical implications. In this context, a method that reliably flags suspicious labels for expert review is itself of significant practical value.
>
> 2. To address the reviewer’s concern and demonstrate both SEI’s generalization beyond medical imaging and its effectiveness under real-world human annotation noise, we have added new experiments on the CIFAR-100N [1] dataset (Appendix J, “Generalization of SEI to Natural Images with Real-World Noise”), with the added text highlighted in blue in the revised manuscript.
>
> On CIFAR-100N, we compare SEI against the same set of noisy label detection baselines used in the main paper. As shown in Table 14 of the revised manuscript, SEI again achieves the best F1 score for mislabeled data detection, outperforming the second-best method by 2.2%. The full comparison is:
>
> | **Method**| **INCV** | **BMM** | **GMM** | **AUM** | **CORES** | **CL** | **SIMIFEAT** | **DEFT** | **ReCoV** | **LEMoN** | **Ours** |
> |:--------|:--------:|:--------:|:--------:|:--------:|:----------:|:------:|:------------:|:--------:|:---------:|:---------:|:--------:|
> | F1 (%) | 59.77 | 63.55 | 63.83 | 74.54 | 38.52 | 67.64 | 79.21 | 75.03 | 67.59 | 78.40 | **81.41** |
>
> These results indicate that SEI is not domain-specific. The signed entropy dynamics remain effective on a standard natural image dataset, and SEI robustly handles real human annotation noise—not only synthetic symmetric or confusion-calibrated noise.
>
> [1] Jiaheng Wei, Zhaowei Zhu, Hao Cheng, Tongliang Liu, Gang Niu, and Yang Liu. Learning with noisy labels revisited: A study using real-world human annotations. In ICLR, 2022.

---

> ### Author Response · Authors · 2025-11-27
>
> > Beyond CLIP, how does SEI perform with standard CNNs/ViTs trained from scratch or with supervised pretraining? Did the authors observe any meaningful diferences in how diferent architectures present diferent entropy values/tendencies in noisy datasets?
>
> Many thanks for the questions. We evaluated SEI on standard classification networks beyond CLIP, including ResNet-50 and ViT.
>
>
> **Table 1:** SEI performance across different backbones on the **ISIC** dataset under varying noise rates.
>
> | **Backbone**       | **0.1** | **0.2** | **0.3** | **0.4** | **0.5** |
> |:------------------:|:-------:|:-------:|:-------:|:-------:|:-------:|
> | RNet-50            | 45.68   | 57.58   | 65.07   | 73.04   | 76.33   |
> | ViT-B/16           | 44.25   | 56.71   | 64.57   | 72.12   | 75.91   |
> | CLIP (RNet-50)     | **47.75** | **59.35** | **67.25** | **74.98** | **79.91** |
> | CLIP (ViT-B/16)    | 46.66   | 58.36   | 66.89   | 73.59   | 77.25   |
>
> **Table 2:** SEI performance across different backbones on the **DeepDRiD** dataset under varying noise rates.
>
> | **Backbone**       | **0.1** | **0.2** | **0.3** | **0.4** | **0.5** |
> |:------------------:|:-------:|:-------:|:-------:|:-------:|:-------:|
> | RNet-50            | 43.99   | 51.34   | 61.17   | 66.89   | 71.75   |
> | ViT-B/16           | 42.88   | 50.47   | 60.71   | 65.68   | 71.31   |
> | CLIP (RNet-50)     | **46.59** | **52.65** | **62.84** | **68.35** | **73.04** |
> | CLIP (ViT-B/16)    | 44.58   | 51.78   | 61.27   | 67.74   | 72.41   |
>
> **Table 3:** SEI performance across different backbones on the **PANDA** dataset under varying noise rates.
>
> | **Backbone**       | **0.1** | **0.2** | **0.3** | **0.4** | **0.5** |
> |:------------------:|:-------:|:-------:|:-------:|:-------:|:-------:|
> | RNet-50            | 71.21   | 75.64   | 79.21   | 79.28   | 78.25   |
> | ViT-B/16           | 69.44   | 74.17   | 78.89   | 78.86   | 76.03   |
> | CLIP (RNet-50)     | **73.17** | **78.21** | **81.86** | **81.96** | **81.85** |
> | CLIP (ViT-B/16)    | 71.92   | 76.52   | 80.63   | 81.28   | 79.45   |
>
>
> SEI maintains competitive mislabeled-sample detection performance on these CNN/ViT models. While the absolute F1 scores are slightly lower than those obtained with CLIP, the relative improvements over competing methods remain consistent. This indicates that the primary benefit comes from the SEI statistic itself, whereas CLIP contributes additional gains through stronger representations and its contrastive training objective. Furthermore, we did not observe meaningful architecture-specific differences in entropy patterns; the trends are consistent across models, as illustrated in Figures 1-4.

---

> ### Author Response · Authors · 2025-11-27
>
> > How sensitive is SEI ranking to training length (early vs late epochs), LR schedules, strong augmentation, label smoothing, and weight decay? Is there a recommended epoch window for a stable ranking?
>
> Thanks a lot for the questions. In Appendix H (“Sensitivity to Training Configuration”), we provide extensive hyperparameter sensitivity studies on the PANDA dataset with 50% noise. These experiments examine training length, learning rate, augmentation, label smoothing, weight decay, and different epoch windows. We also highlight this section in blue for clarity.
>
> **Training length and learning rate.**
>
> On PANDA (noise rate = 0.5), varying the training length from 100 to 200 epochs produces very similar F1 scores (default: 150 epoches):
>
> | **Epochs** | **100** | **150** | **200** | **mean** | **std** |
> |:----------|:-------:|:-------:|:-------:|:--------:|:-------:|
> | F1 (%)     | 81.34   | 81.85   | 81.02   | 81.40    | 0.42    |
>
> The small standard deviation (0.42%) indicates that SEI is insensitive to moderate changes in training duration as long as the model reaches convergence. In contrast, early stopping at epoch 75 leads to a noticeable drop (Table 6), confirming that SEI assumes a reasonably converged model rather than an undertrained one.
>
> For **learning rate**, we sweep from 1e-4 to 1e-2 (default: 1e-3):
>
> | **Learning rate** | **1e-4** | **5e-4** | **1e-3** | **5e-3** | **1e-2** |
> |:-----------------|:--------:|:--------:|:--------:|:--------:|:--------:|
> | F1 (%)            | 80.06    | 81.50    | 81.85    | 78.86    | 77.00    |
>
> SEI is stable in the common LR range for CLIP/ResNet (5e-4–1×1e-3), degrading only when the LR becomes excessively large.
>
> **Data augmentation, label smoothing, and weight decay.**
>
> We also test stronger augmentation and MixUp (default: weak augmentation):
>
> | **Augmentation setting** | **Strong aug.** | **Weak aug.** | **MixUp** |
> |:------------------------|:--------------:|:-------------:|:---------:|
> | F1 (%)                   | 82.38          | 81.85         | 80.88     |
>
> “Strong aug.” adds random affine transforms and random erasing. SEI remains robust under both weak and strong augmentations, with a slight gain under stronger augmentation. MixUp—whose heavy mixing disrupts label–prediction alignment—slightly reduces performance.
>
> For label smoothing (default: 0.0) and weight decay (default: 1e-4):
>
> | **Label smoothing** | **0.0** | **0.1** | **0.2** |
> |:-------------------|:-------:|:-------:|:-------:|
> | F1 (%)              | 81.85   | 81.45   | 80.36   |
>
> Moderate label smoothing slightly degrades performance, and stronger smoothing further weakens entropy signals, as expected.
>
> | **Weight decay** | **5e-5** | **1e-4** | **5e-4** |
> |:----------------|:--------:|:--------:|:--------:|
> | F1 (%)           | 81.36    | 81.85    | 82.21    |
>
> Weight decay within [5e-5, 5e-4] has only mild impact, with slightly higher F1 at the upper end.
>
> **Recommended epoch window.**
>
> To address the “early vs. late epoch” question directly, we compute SEI over sliding 50-epoch windows (step size 20):
>
> | **Window (epochs)** | **[0, 49]** | **[20, 69]** | **[40, 89]** | **[60, 109]** | **[80, 129]** | **[100, 149]** |
> |:-------------------|:-----------:|:------------:|:------------:|:-------------:|:-------------:|:--------------:|
> | F1 (%)              | 75.34       | 50.78        | 49.92        | 51.03         | 54.09         | 56.81          |
>
> All window-based variants perform significantly worse than full-trajectory SEI, consistent with the SEI@Early/SEI@Late results in Table 6. This confirms that SEI benefits from integrating the entire training trajectory rather than relying on a narrow early or late slice.
>
> **In summary**, SEI is robust under standard training-to-convergence settings and typical hyperparameter choices. The ranking is stable with respect to reasonable training lengths, LR schedules, and regularization. Our recommendation is to use the full training trajectory; extreme early stopping or highly atypical hyperparameters are not ideal conditions for SEI.

---

> ### Author Response · Authors · 2025-11-27
>
> > Samples that are intrinsically hard or minority-pattern but correctly labeled may retain higher entropy and risk being flagged as noisy (class imbalance/long-tail scenarios). Do you have analyses showing that SEI does not systematically filter rare but correctly labeled patterns (e.g., minority subtypes)? Any per-class or per-subpopulation error analysis?
>
> We appreciate the reviewer’s concern that intrinsically hard or minority-pattern samples might be incorrectly flagged as noisy. To examine this, we conducted a per-class analysis of noisy label detection performance on PANDA with 50% noise. For each class, we report the class-wise false positive rate (FPR) and per-class F1 score:
>
>
> | **Class**             | **Per-class FPR (%)** | **Per-class F1 (%)** |
> |:---------------------|:---------------------:|:---------------------:|
> | Benign epithelium     | 13.66                 | 85.47                 |
> | Gleason 3             | 22.87                 | 80.38                 |
> | Gleason 4             | 25.87                 | 80.14                 |
> | Gleason 5             | 21.63                 | 82.91                 |
>
> These results show that SEI does not systematically over-filter any particular class. Notably, benign epithelium—the smallest class—has the lowest false positive rate (13.66%), suggesting that minority patterns are not disproportionately removed. Overall, SEI achieves consistently strong detection quality across all classes.
>
> This outcome aligns with the design of SEI: by integrating signed entropy over the entire training trajectory, SEI naturally separates hard-but-correct samples from mislabeled ones. Hard, correctly labeled samples may have high entropy early in training but eventually align with their labels and accumulate positive signed contributions. In contrast, mislabeled samples remain misaligned for most of training and accumulate negative contributions. This reduces the risk of misclassifying intrinsically hard or minority-subtype samples as noisy. We acknowledge that PANDA, while imbalanced, is not an extreme long-tail dataset with ultra-rare subclasses. In the revised manuscript, we include the above per-class analysis in Appendix M (“Per-Class Analysis of Noisy-Label Detection”) and highlight the new text in blue.

---

> ### Author Response · Authors · 2025-11-27
>
> > In which regimes does SEI struggle? Have you explored more extreme noise rates or class imbalances?
>
> Thanks alot for the questions. From a medical perspective, datasets with more than 50% label disagreement are generally considered unreliable for realistic clinical workflows. For this reason, our main experiments focus on noise rates up to 0.5, which already represent a highly challenging yet still meaningful regime for medical imaging.
>
> To directly address the reviewer’s concern, we performed additional experiments on the PANDA dataset with more extreme noise levels (0.6, 0.7, and 0.8) under both confusion-calibrated and symmetric noise. The F1 scores for noisy-label detection are:
>
> | **Noise ratio**        | **0.6** | **0.7** | **0.8** |
> |:----------------------|:-------:|:-------:|:-------:|
> | Confusion-calibrated   | 81.07   | 79.93   | 81.83   |
> | Symmetric              | 81.66   | 80.95   | 80.51   |
>
>
> Even at very high noise rates (60–80%), SEI remains stable around ~80% F1, without a significant performance collapse. Performance does not degrade monotonically; instead, it fluctuates slightly within a narrow band, suggesting that SEI can still capture useful training-dynamics signals even when a large fraction of labels is corrupted. We will include this discussion in Appendix L (“SEI under Extreme Noise Rates”) and highlight the new text in blue.
>
> Regarding class imbalance, PANDA is not the only imbalanced dataset in our study: both ISIC and DeepDRiD also exhibit substantial imbalance, which is typical in medical imaging. Across all three datasets, SEI performs consistently well, indicating robustness to standard levels of class imbalance.
>
> Finally, our hyperparameter sensitivity studies show that SEI becomes more challenging only under clearly unreasonable configurations—such as excessively large learning rates or aggressive early stopping before convergence. Under standard “train-to-convergence with reasonable hyperparameters” settings, SEI remains stable.
>
> Overall, within the practically relevant noise range for medical imaging (0.1–0.5 in the main paper), and even under synthetic noise rates as high as 0.8 in our extended experiments, SEI behaves consistently, and we do not observe clear failure modes in our current evaluations.

---

### Official Review · Reviewer_DQuc · 2025-11-02

**Soundness:** 2
**Presentation:** 3
**Contribution:** 2
**Rating:** 4
**Confidence:** 3

**Summary:**

The paper propose a metric called SEI for identifying mislabeled data, considering both entropy dynamics and label-prediction consistency. The metric is motivated by the observation that correctly labeled samples exhibit consistent entropy decrease during training, while mislabeled samples maintain relatively high entropy throughout training. Experiments on multiple benchmarks shows competitive performance.

**Strengths:**

The approach is simple, architecture-agnostic, and easy to integrate. Experiments demonstrate strong performance across multiple benchmarks.

**Weaknesses:**

my major concern is that the method is build upon empirical observation on a handful of dataset and noise setting. While the presented results seems promising, it is unclear when SEI is expected to succeed or fail.

More discussion and intuition is needed to explain why the proposed SEI works. For example, for high capacity network, the model has the potential to remember the wrong label, this might directly impact the label-prediction alignment pattern (one of the main component in the SEI).

Furthermore, because the experiments rely on controlled, artificially generated noise, it is unclear whether the training-dynamics patterns SEI exploits are specific to these noise models or generalize to real world applications.

**Questions:**

It seems odd to me to frame inter-observer variability as ‘label noise.’ In my view, such variability is more about uncertainty, not necessarily wrong labels. How should SEI handle inter-observer variability?

---

> ### Author Response · Authors · 2025-11-27
>
> > My major concern is that the method is build upon empirical observation on a handful of dataset and noise setting. While the presented results seems promising, it is unclear when SEI is expected to succeed or fail.
>
>  Many thanks for the comment.
>
> Our exposition begins with empirical observations only as a way to build intuition before introducing the formal method. The design of SEI is not based solely on a few isolated cases.
>
> There is theoretical support for using loss or uncertainty/entropy signals to distinguish clean and noisy labels, as shown in prior work [1,2].
>
> In addition, the patterns we exploit are not tied to a particular dataset or noise model. We consistently observe similar signed entropy and alignment dynamics across three distinct medical imaging modalities and tasks (dermoscopy, histopathology, and fundus photography) and under two corruption strategies (symmetric and confusion-calibrated) across a range of noise rates. This breadth of evidence suggests that SEI is not capturing an artifact of a specific dataset or synthetic noise type.
>
> To further clarify when SEI is expected to succeed or degrade, we have added sensitivity studies in Appendix H (“Sensitivity to Training Configuration”, highlighted in blue). On PANDA with 50% noise, varying the number of epochs among 100, 150, and 200 yields almost identical F1 scores (mean 81.40%, std 0.42%), while aggressively early-stopping at epoch 75 reduces performance. Likewise, SEI remains stable for typical learning rates (e.g., 5e-4 and 1e-3), but performance degrades when the learning rate becomes excessively large.
>
> These results show that SEI performs reliably under standard training-to-convergence with reasonable hyperparameters, and that degradation mainly occurs when the underlying classifier is undertrained or trained with clearly suboptimal settings.
>
> [1] Xian-Jin Gui, Wei Wang, and Zhang-Hao Tian. Towards understanding deep learning from noisy labels with small-loss criterion. In IJCAI, 2021.
>
> [2] Daehyun Ji, Dokwan Oh, Yoonsuk Hyun, Oh-Min Kwon, and Myeong-Jin Park. How to handle noisy labels for robust learning from uncertainty. Neural Networks, 143: 209-217, 2021.

---

> ### Author Response · Authors · 2025-11-27
>
> > More discussion and intuition is needed to explain why the proposed SEI works. For example, for high capacity network, the model has the potential to remember the wrong label, this might directly impact the label-prediction alignment pattern (one of the main component in the SEI).
>
> We appreciate the reviewer’s thoughtful comment. The concern about high-capacity networks memorizing wrong labels typically applies to the **final converged** model. In contrast, SEI is computed by recording the signed entropy at every training step and integrating over the entire training trajectory. Even if some mislabeled samples are eventually memorized near the end of training, these late aligned epochs behave as outliers and are outweighed by the much longer period during which the model disagrees with the incorrect label. This temporal integration is precisely what makes SEI robust to memorization effects.
>
> Our ablation studies (Section 4.3.2 and Appendix F.2) support this behavior: SEI performs best when integrating over the full trajectory, whereas early-only or late-only variants are less effective. This confirms that SEI works because the discriminative patterns appear consistently throughout training, not just at the end.
>
> We acknowledge that this intuition was not sufficiently emphasized in the original submission. We now include a short discussion in Appendix H (“Temporal integration window”, highlighted in blue) and clarify that SEI assumes a standard training-to-convergence regime. We also note that insufficient training or overly aggressive overtraining may weaken the signal—an explicit limitation we now state in the revised version.

---

> ### Author Response · Authors · 2025-11-27
>
> > Furthermore, because the experiments rely on controlled, artificially generated noise, it is unclear whether the training-dynamics patterns SEI exploits are specific to these noise models or generalize to real world applications.
>
> We thank the reviewer for this important question. To assess the generalization ability of SEI beyond the medical domain and to evaluate its behavior under real-world human annotation noise, we conducted additional experiments on the CIFAR-100N [1] dataset. These results are included in Appendix J (“Generalization of SEI to Natural Images with Real-World Noise”) and highlighted in blue in the revised manuscript.
>
> On CIFAR-100N, we compare SEI against the same set of noisy label detection baselines used in the main paper. As shown in Table 14 of the revised manuscript, SEI again achieves the best F1 score for mislabeled data detection, outperforming the second-best method by 2.2%. The full comparison is:
>
> | **Method**| **INCV** | **BMM** | **GMM** | **AUM** | **CORES** | **CL** | **SIMIFEAT** | **DEFT** | **ReCoV** | **LEMoN** | **Ours** |
> |:--------|:--------:|:--------:|:--------:|:--------:|:----------:|:------:|:------------:|:--------:|:---------:|:---------:|:--------:|
> | F1 (%) | 59.77 | 63.55 | 63.83 | 74.54 | 38.52 | 67.64 | 79.21 | 75.03 | 67.59 | 78.40 | **81.41** |
>
> These results indicate that SEI is not domain-specific. The signed entropy dynamics remain effective on a standard natural image dataset, and SEI robustly handles real human annotation noise—not only synthetic symmetric or confusion-calibrated noise.
>
> [1] Jiaheng Wei, Zhaowei Zhu, Hao Cheng, Tongliang Liu, Gang Niu, and Yang Liu. Learning with noisy labels revisited: A study using real-world human annotations. In ICLR, 2022.

---

> ### Author Response · Authors · 2025-11-27
>
> > It seems odd to me to frame inter-observer variability as ‘label noise.’ In my view, such variability is more about uncertainty, not necessarily wrong labels. How should SEI handle inter-observer variability?
>
> We thank the reviewer for this valuable comment. Inter-observer variability and label noise are related but not identical. Variability across annotators often reflects inherent uncertainty, whereas noisy label detection assumes a single hard label per sample and seeks to identify cases where that label is incorrect. Learning with uncertain or probabilistic labels—such as modeling annotator distributions or ambiguity—is an important but distinct problem setting [1] and is not the focus of this work.
>
> Our paper operates strictly under the standard hard-label noise detection setting, where each training sample is associated with one label, and the goal is to detect mislabeled instances under this assumption. SEI is therefore designed and evaluated within this framework.
>
> This conceptual distinction was not sufficiently clear in the original submission. In Appendix K (highlighted in blue), we now explicitly discuss the relationship between uncertainty and label noise and clarify that our experiments are conducted under the hard-label setting, while learning from uncertain or probabilistic labels is closely related but outside the scope of this paper.
>
> [1] Simon A. A. Kohl, Bernardino Romera-Paredes, Clemens Meyer, Jeffrey De Fauw, Joseph R. Ledsam, Klaus H. Maier-Hein, S. M. Ali Eslami, Danilo Jimenez Rezende, and Olaf Ronneberger. A probabilistic U-Net for segmentation of ambiguous images. In NeurIPS, 2018.

---

### Official Review · Reviewer_tyJm · 2025-11-02

**Soundness:** 3
**Presentation:** 3
**Contribution:** 3
**Rating:** 8
**Confidence:** 3

**Summary:**

This paper proposes a simple yet effective method, Signed Entropy Integral (SEI), for detecting mislabeled samples in training datasets through analysis of training dynamics.
The core insight is that correctly labeled samples exhibit a steady decrease in prediction entropy during training, whereas mislabeled samples maintain persistently high entropy due to model–label inconsistency.

SEI functions as an automatic, architecture-agnostic indicator of label noise.
A self-calibrating threshold is introduced, leveraging auxiliary-class pseudo-errors to distinguish clean from mislabeled samples without requiring external supervision.

Experiments conducted on three medical imaging datasets demonstrate that SEI achieves state-of-the-art performance for mislabeled data detection under both symmetric and confusion-calibrated noise conditions.

Overall, the paper presents a clear and well-motivated study with good empirical support.  It provides valuable insight into leveraging entropy trajectories for robust label noise detection, supported by transparent presentation and convincing experimental validation.

**Strengths:**

**1. Clear Motivation and Conceptual Simplicity**

The paper addresses a well-motivated and underexplored problem—detecting mislabeled samples through training dynamics—and proposes a simple yet effective solution with clear conceptual grounding.

---

**2. Architecture-Agnostic Design and Practical Usability**

SEI operates without architectural modifications and shows strong generalization across both CNN (ResNet) and Transformer (CLIP) backbones.
Its plug-and-play nature and minimal computational cost make it practical for real-world noisy-label scenarios.

---

**3. Interpretability and Diagnostic Insight**

The method provides an intuitive and interpretable perspective on model behavior under label noise.
By linking entropy evolution to label correctness, it offers diagnostic insight into how neural networks react to mislabels during training.

**Weaknesses:**

**1. Sensitivity to Training Configuration**

Since SEI depends on entropy evolution across training, its stability under different optimization schedules remains unclear.
Variations in training duration, learning rate, or early stopping may affect the computed SEI values, raising concerns about reproducibility.

---

**2. Lack of Statistical Significance Analysis**

Although the reported results show consistent improvements, no variance measures or statistical tests are provided.
Without reporting standard deviations or confidence intervals, it is difficult to determine whether the gains are statistically meaningful.

---

**3. Lack of discussion on the stability and generality of the thresholding strategy.**

The auxiliary-class–based threshold still depends on a manually defined sampling ratio \( N / (K + 1) \).
Its robustness under data imbalance, limited samples, or multimodal settings has not been validated.

---

**4. Limited Domain Generalization**

All experiments are conducted on medical imaging datasets.
It remains uncertain whether SEI’s entropy patterns generalize to other domains such as natural image or text classification, where label noise and model calibration behave differently.

**Questions:**

**1. Sensitivity to Training Configuration**

It is unclear how sensitive SEI is to training hyperparameters such as the number of epochs or the learning rate.
Since the method relies on entropy trajectories over training, the stability of SEI under different optimization schedules (e.g., early stopping or extended training) should be clarified.

> **Question:**
> How sensitive is SEI to training duration and learning rate choices?

---

**2. Lack of Statistical Significance Analysis**

Although the experimental tables are comprehensive, they lack statistical tests such as confidence intervals or t-tests.
Without measures of variability, it is difficult to judge whether the observed performance improvements are statistically significant or within the range of random fluctuation.

> **Questions:**
> - Could the authors report standard deviations or statistical tests (e.g., t-test, CI) to validate the robustness of SEI’s improvements?
> - How consistent are the results across multiple random seeds?

---

**3. Stability of the Thresholding Strategy**

The proposed auxiliary-class–based adaptive threshold depends on the ratio \( N / (K + 1) \), which remains a manually defined proportion.
It is unclear whether this strategy is stable under data imbalance or when applied to multimodal tasks.

> **Questions:**
> - How sensitive is the SEI threshold to the number or sampling ratio of auxiliary-class samples?
> - Have the authors tested how performance changes when this ratio is varied?

---

**4. Domain Generalization of the Entropy Trend**

The paper primarily focuses on medical imaging datasets.
However, it remains uncertain whether the same entropy evolution patterns hold in other domains such as natural images, where model calibration and noise characteristics differ.

> **Question:**
> For non-medical tasks, would SEI exhibit similar entropy trajectories, or is this behavior domain-specific?

---

> ### Author Response · Authors · 2025-11-27
>
> > How sensitive is SEI to training duration and learning rate choices?
>
> We thank the reviewer for the insightful question regarding SEI’s sensitivity to training configurations. We conducted additional experiments on the PANDA dataset under the confusion-calibrated 50% noise setting to systematically evaluate the effects of training duration and learning rate. The results show that SEI is highly stable under reasonable training configurations and does not introduce extra hyperparameter fragility beyond that of the underlying classifier.
>
> We first varied the number of training epochs while keeping all other settings fixed (default: 150 epoches):
>
>
> | **Epochs** | **100** | **150** | **200** | **mean** | **std** |
> |:----------|:-------:|:-------:|:-------:|:--------:|:-------:|
> | F1 (%)     | 81.34   | 81.85   | 81.02   | 81.40    | 0.42    |
>
>
> The mean F1 is 81.40% with a standard deviation of only 0.42%, indicating that SEI is insensitive to moderate changes in training duration as long as the model is trained to convergence.
>
> We also applied early stopping at epoch 75, where the F1 score drops to 76.11 (Table 6). This confirms that if training is stopped prematurely—before entropy trajectories fully develop—SEI becomes less effective. This behavior is consistent with our design, which relies on full-trajectory integration. In practice, SEI only assumes a standard, complete training run and does not require any special scheduling.
>
> We then swept the initial learning rate from 1e-4 to 1e-2 (150 epochs, default learning rate 1e-3, all other settings fixed):
>
>
> | **Learning rate** | **1e-4** | **5e-4** | **1e-3** | **5e-3** | **1e-2** |
> |:-----------------|:--------:|:--------:|:--------:|:--------:|:--------:|
> | F1 (%)            | 80.06    | 81.50    | 81.85    | 78.86    | 77.00    |
>
>
> Performance only degrades when the learning rate is excessively large (5e-3, 1e-2), while SEI remains very stable within the range typically used for training CLIP and standard classifiers (5e-4–1e-3). This trend mirrors standard classification training, where an appropriate learning rate is needed for effective learning of the base model itself.
>
> In summary, SEI does not introduce additional sensitivity beyond standard training considerations and remains robust under reasonable hyperparameter choices. This further supports its plug-and-play nature: SEI can be directly integrated into existing training pipelines without modifying the underlying optimization procedure.
>
> We have added these sensitivity studies and the corresponding discussion to Appendix H (“Sensitivity to Training Configuration”) and highlighted the new text in blue in the revised manuscript.

---

> ### Author Response · Authors · 2025-11-27
>
> > Could the authors report standard deviations or statistical tests (e.g., t-test, CI) to validate the robustness of SEI’s improvements? How consistent are the results across multiple random seeds?
>
> We appreciate the reviewer’s suggestion to include statistical evidence and multi-seed results. Following this recommendation, we conducted additional experiments on the PANDA dataset with confusion-calibrated noise at a noise rate of η = 0.5. In addition to our original seed (1), we evaluated SEI using four more seeds: 42, 426, 1005, and 1016. The resulting F1 scores are:
>
>
> | **Seed** | **1** | **42** | **426** | **1005** | **1016** | **mean** | **std** |
> |:--------|:-----:|:------:|:-------:|:--------:|:--------:|:-------:|:------:|
> | F1 (%)   | 81.85 | 82.09  | 82.72   | 81.01    | 81.48    | 81.83   | 0.64   |
>
>
> The results are highly consistent across seeds, with a mean of 81.83% and a standard deviation of only 0.64%, demonstrating that SEI is robust to randomness in initialization and data shuffling.
>
> To further assess statistical significance, we evaluated SEI and the second-best method using the same set of seeds and performed a paired t-test on their F1 scores. SEI shows an average improvement of 5.45% over the second-best baseline, with a 95% confidence interval of [4.5%, 6.1%] for the difference in F1. The paired t-test yields a p-value < 0.01, confirming that the improvement is statistically significant and not attributable to random seed variation.

---

> ### Author Response · Authors · 2025-11-27
>
> > How sensitive is the SEI threshold to the number or sampling ratio of auxiliary-class samples? Have the authors tested how performance changes when this ratio is varied?
>
> We appreciate the reviewer’s concern regarding the robustness of the auxiliary-class–based threshold. To evaluate this, we performed a sensitivity analysis on the PANDA dataset with 50% noise by varying the sampling ratio of auxiliary-class samples. In our default setting, the number of auxiliary samples is N/(K+1). We scaled this number using factors of 0.5×, 0.75×, 1.0×, 1.5×, and 2.0×. The results are:
>
> |  | **0.5×** | **0.75×** | **1.0×** | **1.5×** | **2.0×** | **mean** | **std** |
> |:--------|:-------:|:--------:|:-------:|:-------:|:-------:|:------:|:------:|
> | F1 (%)   | 81.94 | 83.01 | 81.85 | 81.41 | 81.12 | 81.87 | 0.72 |
>
>
> The F1 scores remain within a tight range, with a mean of 81.87% and a standard deviation of only 0.72%, even when the number of auxiliary samples varies by a factor of four. This indicates that SEI is not sensitive to the exact sampling ratio: as long as a reasonable number of auxiliary samples is used, the estimated mean SEI for the auxiliary class remains stable.
>
> It is also worth noting that although the sampling ratio is manually specified, the threshold itself is entirely data-driven and learned adaptively. Moreover, PANDA is intrinsically class-imbalanced, yet the threshold remains stable in this challenging setting.
>
> We have added this sensitivity study and the corresponding discussion to Appendix I (“Sensitivity to the Auxiliary-Class Sampling Ratio”) and highlighted the new text in blue in the revised manuscript.

---

> ### Author Response · Authors · 2025-11-27
>
> > For non-medical tasks, would SEI exhibit similar entropy trajectories, or is this behavior domain-specific?
>
> We thank the reviewer for this important question. To assess the generalization ability of SEI beyond the medical domain and to evaluate its behavior under real-world human annotation noise, we conducted additional experiments on the CIFAR-100N [1] dataset. These results are included in Appendix J (“Generalization of SEI to Natural Images with Real-World Noise”) and highlighted in blue in the revised manuscript.
>
> On CIFAR-100N, we compare SEI against the same set of noisy label detection baselines used in the main paper. As shown in Table 14 of the revised manuscript, SEI again achieves the best F1 score for mislabeled data detection, outperforming the second-best method by 2.2%. The full comparison is:
>
> | **Method**| **INCV** | **BMM** | **GMM** | **AUM** | **CORES** | **CL** | **SIMIFEAT** | **DEFT** | **ReCoV** | **LEMoN** | **Ours** |
> |:--------|:--------:|:--------:|:--------:|:--------:|:----------:|:------:|:------------:|:--------:|:---------:|:---------:|:--------:|
> | F1 (%) | 59.77 | 63.55 | 63.83 | 74.54 | 38.52 | 67.64 | 79.21 | 75.03 | 67.59 | 78.40 | **81.41** |
>
>
> These results indicate that SEI is not domain-specific. The signed entropy dynamics remain effective on a standard natural image dataset, and SEI robustly handles real human annotation noise—not only synthetic symmetric or confusion-calibrated noise.
>
> [1] Jiaheng Wei, Zhaowei Zhu, Hao Cheng, Tongliang Liu, Gang Niu, and Yang Liu. Learning with noisy labels revisited: A study using real-world human annotations. In ICLR, 2022.

---

### Comment · Area_Chair_BiwS · 2025-11-25
**Post-Rebuttal Discussion**

Dear all,
Could you first review the original comments from other reviewers and the rebuttal materials, and then post your comments? Discussion is necessary for this paper.

Best,
AC

---

### Author Response · Authors · 2025-12-03
**A Summary of Our Interactions with Reviewers**

We sincerely thank all Reviewers, ACs, and SACs for their thoughtful and constructive feedback. Below is a concise summary of our interactions.

***

`Reviewer tyJm`

**Main concerns:** (1) Sensitivity to training configurations, (2) statistical significance, (3) stability of the thresholding strategy, (4) domain generalization of the entropy trend.

**Our response:**

(1) We added extensive hyperparameter sensitivity studies (Appendix H).

(2) Multi-seed experiments and a paired t-test show that SEI’s improvements are statistically significant.

(3) We added “Sensitivity to the Auxiliary-Class Sampling Ratio” (Appendix I).

(4) We conducted additional experiments on CIFAR-100N, a real-world natural image dataset (Appendix J).

***

`Reviewer DQuc`

**Main concerns:** (1) When SEI is expected to succeed or fail, (2) why SEI works despite high-capacity networks memorizing wrong labels, (3) generalization to real-world noisy labels, (4) inter-observer variability vs. label noise.

**Our response:**

(1) Appendix H shows that SEI is stable under standard, well-tuned training, and mainly degrades only when the classifier is undertrained or badly configured.

(2) We clarified that SEI remains robust because full-trajectory integration dilutes brief late-stage memorization of wrong labels.

(3) We conducted additional experiments on CIFAR-100N, a real-world natural image dataset (Appendix J).

(4) We added Appendix K to clarify that inter-observer variability and label noise are related but not identical settings.

***

`Reviewer GH2B`

**Main concerns:** (1) Limited novelty relative to prior entropy-based signals, (2) generalization to natural images with real-world noise, (3) sensitivity to training configurations, (4) performance on CNNs/ViTs beyond CLIP, (5) per-class or subpopulation error analysis, (6) performance under extreme noise rates.

**Our response:**

(1) Compared with prior entropy-based signals, SEI introduces the novel concept of signed entropy and further leverages full-trajectory integration and an adaptive thresholding strategy. Together, these make it a method that goes beyond existing entropy-based approaches.

(2) We added experiments on CIFAR-100N, a real-world natural image dataset (Appendix J).

(3) We included extensive hyperparameter sensitivity analyses (Appendix H).

(4) Section “Architecture Generalizability” and Table 4 show SEI is robust across architectures.

(5) Appendix M now includes class-wise FPR and F1, showing SEI does not systematically over-filter any class.

(6) Appendix L demonstrates that SEI behaves consistently even under extreme noise levels (0.6, 0.7, 0.8).

***

`Reviewer kas8`

**Main concerns:** (1) Generalization to natural images with real-world noise, (2) comparison to entropy-based baselines (AAAI 2023, Neurocomputing 2025), (3) SEI’s computational overhead, (4) thresholding heuristics and dataset-specific tuning.

**Our response:**

(1) Additional experiments on CIFAR-100N, a real-world natural image dataset, were added (Appendix J).

(2) We explained why the suggested methods cannot be directly compared to ours.

(3) We analyzed and quantitatively showed that SEI introduces negligible computational overhead.

(4) Appendix I demonstrates that SEI is insensitive to the exact sampling ratio and that the thresholding mechanism is data-driven rather than tuned per dataset.

***

**All updates are highlighted in blue in the revised manuscript.**

---

### Meta-Review · Area_Chair_SSiE · 2026-01-06

**Summary:**

This paper introduces a new in-training metric for identifying mislabeled samples based on the entropy of the softmax probabilities and the correctness of the prediction. While the idea is clear and the experiments suggest strong empirical performance, there are several outstanding weaknesses that make the paper unsuitable for ICLR in its current form:

* The method is very similar to prior work that uses entropy, margin, loss, or other related metrics during training to identify mislabeled data. The authors should provide additional analysis to explain why their specific design decisions are necessary to improve performance (i.e., why signed entropy is better than the margin) and to demonstrate the significance of their work.
* Most of the experiments are limited to synthetic noise distributions. The authors added a real-world noise dataset in the rebuttal, but they still lack common benchmarks such as Clothing1M and WebVision.
* The paper’s scope is unnecessarily narrow: focusing on medical images and fine-tuned CLIP classifiers. While the authors extended these results to more general image datasets/models in the rebuttal, the results feel largely incomplete and could benefit from greater breadth.

Moreover, I would encourage the authors to explicitly note the similarities between their method and the Area Under the Margin (AUM) method. Two of the major methodological components of the paper - (1) integrating across training epochs and (2) using threshold samples to determine the clean/mislabeled cutoff - were first introduced in the AUM paper, but the authors make no acknowledgement of this connection in their paper. I assume that the authors are aware of this prior work, given that they cite and compare against AUM, and so it is hard to read this lack of attribution as anything but academic dishonesty.

**Reviewer Concerns:**

Because none of the reviewers engaged at all, I had to use my best judgment to determine which concerns/questions had been resolved.

Resolved concerns:

* Questions about sensitivity to hyperparameters, rare cases, and settings where the proposed method fails.
* Statistical significance analysis
* Details on the thresholding strategy

Unresolved concerns:

* Relation to prior work
* Limited scope and non-synthetic datasets
* Analysis experiments to understand the method better.

**Reviewer Scores:**

Because none of the reviewers engaged at all, I do not think that the reviewers would have engaged with this paper and changed their scores even if the discussion period had not been cut short. I acknowledge that this outcome is unfortunate and a poor reflection on the reviewers, but the outcome is no more unfair than it would have been under a normal reviewing period.

---

### Decision · Program_Chairs · 2026-01-26

Reject